# DSPy: Compiling Declarative Language Model Calls into Self-Improving Pipelines

**Omar Khattab,**[1] **Arnav Singhvi,**[2]
**Paridhi Maheshwari,**[4] **Zhiyuan Zhang,**[1]
**Keshav Santhanam,**[1] **Sri Vardhamanan,**[5] **Saiful Haq,**[6]
**Ashutosh Sharma,**[6] **Thomas T. Joshi,**[7] **Hanna Moazam,**[8]
**Heather Miller,**[3,9] **Matei Zaharia,**[2] **Christopher Potts**[1]

[1]Stanford University, [2]UC Berkeley, [3]Carnegie Mellon University,
[4]Amazon Alexa AI, [5]Dashworks Technologies, Inc., [6]IIT Bombay,
[7]Calera Capital, [8]Microsoft Qatar, [9]Two Sigma Investments

okhattab@cs.stanford.edu

## Abstract

The ML community is rapidly exploring techniques for prompting language models (LMs) and for stacking them into pipelines that solve complex tasks. Unfortunately, existing LM pipelines are typically implemented using hard-coded "prompt templates", i.e. lengthy strings discovered via trial and error. Toward a more systematic approach for developing and optimizing LM pipelines, we introduce DSPy, a programming model that abstracts LM pipelines as *text transformation graphs*, or imperative computational graphs where LMs are invoked through *declarative* modules. DSPy modules are *parameterized*, meaning they can learn how to apply compositions of prompting, finetuning, augmentation, and reasoning techniques. We design a compiler that will optimize any DSPy pipeline to maximize a given metric, by creating and collecting demonstrations. We conduct two case studies, showing that succinct DSPy programs can express and optimize pipelines that reason about math word problems, tackle multi-hop retrieval, answer complex questions, and control agent loops. Within minutes of compiling, DSPy can automatically produce pipelines that outperform out-of-the-box few-shot prompting as well as expert-created demonstrations for GPT-3.5 and Llama2-13b-chat. On top of that, DSPy programs compiled for relatively small LMs like 770M parameter T5 and Llama2-13b-chat are competitive with many approaches that rely on large and proprietary LMs like GPT-3.5 and on expert-written prompt chains. DSPy is available at https://github.com/stanfordnlp/dspy.

## 1 Introduction

Language models (LMs) are enabling researchers to build NLP systems at higher levels of abstraction and with lower data requirements than ever before (Bommasani et al., 2021). This is fueling an exploding space of "prompting" techniques—and lightweight finetuning techniques—for *adapting* LMs to new tasks (Kojima et al., 2022), eliciting systematic *reasoning* from them (Wei et al., 2022; Wang et al., 2022b), and *augmenting* them with retrieved sources (Guu et al., 2020; Lazaridou et al., 2022) or with tools (Yao et al., 2022). Most of these techniques are explored in isolation, but interest has been growing in building multi-stage *pipelines* and *agents* that decompose complex tasks into more manageable calls to LMs in an effort to improve performance (Qi et al., 2019; Khattab et al., 2021a; Karpas et al., 2022; Dohan et al., 2022; Pourreza & Rafiei, 2023; Yao et al., 2023).

Unfortunately, LMs are known to be sensitive to how they are prompted for each task, and this is exacerbated in pipelines where multiple LM calls have to *interact* effectively. As a result, the LM calls in existing LM pipelines and in popular developer frameworks are generally implemented using hard-coded 'prompt templates', that is, long strings of instructions and demonstrations that are hand crafted through manual trial and error. We argue that this approach, while pervasive, can be brittle

and unscalable—conceptually akin to hand-tuning the weights for a classifier. A given string prompt might not generalize to different pipelines or across different LMs, data domains, or even inputs.

Toward a more systematic approach to designing AI pipelines, we introduce the **DSPy** programming model.[1] DSPy pushes building new LM pipelines away from manipulating free-form strings and closer to *programming* (composing modular operators to build text transformation graphs) where a compiler automatically generates optimized LM invocation strategies and prompts from a program. We draw inspiration from the consensus that emerged around neural network abstractions (Bergstra et al., 2013), where (1) many general-purpose layers can be modularly *composed* in any complex architecture and (2) the model weights can be *trained* using optimizers instead of being hand-tuned.

To this end, we propose the **DSPy programming model** (Section 3). We first translate string-based prompting techniques, including complex and task-dependent ones like Chain of Thought (Wei et al., 2022) and ReAct (Yao et al., 2022), into declarative modules that carry *natural-language typed signatures*. DSPy modules are task-adaptive components—akin to neural network layers—that abstract any particular text transformation, like answering a question or summarizing a paper. We then parameterize each module so that it can *learn* its desired behavior by iteratively bootstrapping useful demonstrations within the pipeline. Inspired directly by PyTorch abstractions (Paszke et al., 2019), DSPy modules are used via *define-by-run* computational graphs. Pipelines are expressed by (1) declaring the modules needed and (2) using these modules in any logical control flow (e.g., `if` statements, `for` loops, exceptions, etc.) to logically connect the modules.

We then develop the **DSPy compiler** (Section 4), which optimizes any DSPy program to improve quality or cost. The compiler inputs are the program, a few training inputs with optional labels, and a validation metric. The compiler simulates versions of the program on the inputs and *bootstraps* example traces of each module for self-improvement, using them to construct effective few-shot prompts or finetuning small LMs for steps of the pipeline. Optimization in DSPy is highly modular: it is conducted by *teleprompters*,[2] which are general-purpose optimization strategies that determine how the modules should learn from data. In this way, the compiler automatically maps the declarative modules to *high-quality* compositions of prompting, finetuning, reasoning, and augmentation.

Programming models like DSPy could be assessed along many dimensions; we focus on the role of hand-crafted prompts in shaping system performance. We are seeking to reduce or even remove their role through DSPy modules and teleprompters. We report on two expansive case studies: math word problems (GSM8K; Cobbe et al. 2021) and multi-hop question answering (HotPotQA; Yang et al. 2018) with explorations of chain of thought, multi-chain reflection, multi-hop retrieval, retrieval-augmented question answering, and agent loops. Our evaluations use a number of different compiling strategies effectively and show that straightforward DSPy programs outperform systems using hand-crafted prompts, while also allowing our programs to use much smaller and hence more efficient LMs effectively. Overall, the main contributions of this paper are empirical and algorithmic: with DSPy, we have found that we can implement very short programs that can bootstrap state-of-the-art multi-stage NLP systems using LMs as small as Llama2-13b with no hand-crafted prompts.

## 2 RELATED WORK

This work is inspired by the role that Torch (Collobert et al., 2002), Theano (Bergstra et al., 2010; 2011; Al-Rfou et al., 2016), Chainer (Tokui et al., 2015), and others played in the development in deep learning by providing powerful abstractions. We are seeking to offer a solid conceptual framework and programming abstractions for what we call *foundation model programming*. We draw on differentiable programming (Wang et al., 2018) but applied to LM calls rather than neural networks, and borrow syntactic elements from PyTorch (Paszke et al., 2019).

In-context learning (McCann et al. 2018; Radford et al. 2018; Brown et al. 2020) is a key mechanism for programming LMs in this new mode. A growing body of work has revealed that, especially with instruction tuning (Ouyang et al., 2022), we can elicit sophisticated behavior via prompting (Wei et al., 2022; Wang et al., 2022b; Press et al., 2022; Yao et al., 2022; Khot et al., 2022; Madaan et al., 2023). Similarly, forms of weak supervision that would normally require task-specific (Khattab

---

[1] DSPy is pronounced *dee-ess-pie*. It's the second iteration of our earlier Demonstrate–Search–Predict framework (DSP; Khattab et al. 2022).

[2] We derive the name *tele*-prompters from the notion of abstracting and automating the task of prompting—such it happens *at a distance*, without manual intervention.

et al., 2021a;b) or hand-built (Ratner et al., 2016; Hancock et al., 2018) heuristics are now done by LMs (Wang et al., 2022b; Zelikman et al., 2022; Zhang et al., 2022; Shao et al., 2023). In-context learning methods now routinely invoke tools, leading to LM pipelines that use retrieval (Chen et al., 2017; Lewis et al., 2020; Guu et al., 2020; Lazaridou et al., 2022; Izacard et al., 2022), and APIs (Nakano et al., 2021). Toolkits have been developed to facilitate this, including LangChain (Chase, 2022), Semantic Kernel (Microsoft, 2023), and LlamaIndex (Liu, 2022), and many other retrieval or agent libraries. While these toolkits provide pre-packaged chains and agents that connect LMs with tools, they suffer from the pervasive prompt engineering challenges we address in DSPy: in particular, their abstractions still express task-specific behavior through hand-written prompt templates (see Appendix D). LMQL (Beurer-Kellner et al., 2023) is a related query language, which efficiently constrains the decoding algorithms of LMs to generate only strings that fulfill logical constrains (e.g., lists of bullets or values formatted correctly for a calculator). Whereas DSPy focuses on optimizing LM pipelines for a given metric, LMQL's lower-level interface for controlled decoding could be beneficial to implement specific advanced modules within DSPy.

Researchers are starting to apply discrete optimization and RL to find effective prompts, generally for a single logical LM call (Guo et al., 2023; Pryzant et al., 2023; Huang et al., 2022; Yang et al., 2023). DSPy seeks to generalize this space: it offers a rich framework for optimizing *arbitrary pipelines* from *high-level declarative signatures*, by bootstrapping high-quality *multi-stage demonstrations* with constraints. In this framework, DSPy teleprompters may apply optimization using model selection techniques like cross-validation or, in principle, with sophisticated techniques involving RL and LM feedback (Hu et al., 2023; Zhao et al., 2023a; Shinn et al., 2023) or learned or Bayesian hyperparameter optimization methods (Bergstra et al., 2013; Akiba et al., 2019). The present paper seeks to motivate DSPy as a programming model and to report new empirical findings from applying the DSPy compiler. This is inspired by formative work by Bergstra et al. (2010; 2013), Paszke et al. (2019), and Wolf et al. (2020), who support their respective programming models with a mix of benchmark numbers and some qualitative measures. For the current paper, we focus on showing that DSPy and its compiler allow us to build outstanding LM systems without hand-crafted prompt strings, but instead from truly modular units, and that this opens up doors for systematically exploring a rich design space at a very high programmatic level of abstraction.

## 3 THE DSPy PROGRAMMING MODEL

We present DSPy, which treats LMs as abstract devices for text generation,[3] and optimizes their usage in arbitrary computational graphs. DSPy programs are expressed in Python: each program takes the task input (e.g., a question to answer or a paper to summarize) and returns the output (e.g., an answer or a summary) after a series of steps. DSPy contributes three abstractions toward automatic optimization: signatures, modules, and teleprompters. Signatures abstract the input/output behavior of a module; modules replace existing hand-prompting techniques and can be composed in arbitrary pipelines; and teleprompters optimize each module in the pipeline to maximize a metric.

### 3.1 NATURAL LANGUAGE SIGNATURES CAN ABSTRACT PROMPTING & FINETUNING

Instead of free-form string prompts, DSPy programs use *signatures* to assign work to the LM. This is a *natural-language typed* function declaration: a declarative spec that tells DSPy *what* a transformation needs to do (e.g., consume questions and return answers), rather than *how* a specific LM should be prompted to implement that behavior. More formally, a DSPy signature is a tuple of *input fields* and *output fields* (and optional *instruction*). A field consists of *field name* and optional metadata.[4] The roles of fields are inferred from field names, e.g. the compiler will use in-context learning to interpret question differently from answer and to iteratively refine its usage of these fields.

Signatures offer two benefits over prompts: they support compilation into high-quality, self-improving, and pipeline-adaptive prompts or finetunes. This is primarily done by bootstrapping (Sec 4) useful demonstrating examples for each signature. Additionally, they handle structured formatting and parsing logic to reduce (or, ideally, avoid) brittle string manipulation in user programs.

---

[3]We assume access to one or more LMs, which consume a prompt string and return text completions. This may be a promptable LM capable of in-context learning (e.g., GPT-3.5 or Llama2-7b) or a smaller finetuneable LM (e.g., T5-base). An LM may be selected as the default; operations will use it unless configured otherwise.

[4]Description of the task is optional and usually omitted. Fields can carry optional *prefix* and *description*.

In practice, DSPy signatures can be expressed with a shorthand notation like `question -> answer`, so that line 1 in the following is a complete DSPy program for a basic question-answering system (with line 2 illustrating usage and line 3 the response when GPT-3.5 is the LM):

```
1 qa = dspy.Predict("question -> answer")
2 qa(question="Where is Guaraní spoken?") # Out: Prediction(answer='Mainly in South America.')
```

In the shorthand notation, each field's name indicates the semantic role that the input (or output) field plays in the transformation. DSPy will parse this notation and expand the field names into meaningful instructions for the LM, so that `english_document -> french_translation` would prompt for English to French translation. When needed, DSPy offers more advanced programming interfaces for expressing more explicit constraints on signatures (Appendix B).

## 3.2 PARAMETERIZED & TEMPLATED MODULES CAN ABSTRACT PROMPTING TECHNIQUES

Akin to type signatures in programming languages, DSPy signatures define an interface and provide type-like hints on the expected behavior. To use a signature, we declare a *module* with that signature, like the `Predict` module above. A module declaration like this returns a *function* having that signature. The core module in DSPy is `Predict` (Appendix G.1), which stores internally the supplied signature, an optional LM to use (initially `None`, but otherwise overrides the default LM for this module), and a list of demonstrations for prompting (initially empty). The instantiated module behaves as a callable function: it takes in keyword arguments corresponding to the signature input fields (e.g., `question`), formats a prompt to implement the signature and include the appropriate demonstrations, calls the LM, and parses the output fields. When `Predict` detects it's being used in `compile` mode, it internally tracks input/output traces to assist bootstrapping.

DSPy includes more sophisticated modules like `ChainOfThought`, `ProgramOfThought`, `MultiChainComparison`, and `ReAct`.[5] These can be used interchangeably to implement a DSPy signature, e.g. changing `Predict` to `ChainOfThought` above leads to a system that thinks step by step before committing to its output field. Importantly, these modules are implemented in a few lines of code by expanding the user-defined signature and calling `Predict` one or more times on new signatures. We show a simplified implementation of `ChainOfThought` in Appendix G.2 using seven lines of code. This is a fully-fledged module capable of learning effective few-shot prompting for any LM or task. We contrast that with Appendix E, which copies long reasoning prompts hand-written by recent papers and prompting libraries.

DSPy modules translate prompting techniques into modular functions that support any signature by *parameterizing* these modules. To understand this, observe that any LM call seeking to implement a particular signature needs to specify *parameters* that include: (1) the specific LM to call (Chen et al., 2023), (2) the prompt instructions (Yang et al., 2023) and the string prefix of each signature field and, most importantly, (3) the demonstrations used as few-shot prompts (for frozen LMs) or as training data (for finetuning). We focus in this paper on generating and selecting useful demonstrations and using them for prompting or finetuning. We find that bootstrapping demonstrations gives us a powerful way to teach sophisticated pipelines of LMs new behaviors systematically.

DSPy modules can be composed in arbitrary pipelines in a define-by-run interface. Inspired directly by PyTorch and Chainer, one first declares the modules needed at initialization, allowing DSPy to keep track of them for optimization, and then one expresses the pipeline with arbitrary code that calls the modules in a `forward` method. As a simple illustration, we offer the following simple but complete retrieval-augmented generation (RAG) system.[6] DSPy programs may use tools, which are modules that execute computation. We support retrieval models through a `dspy.Retrieve` module. At the time of writing, DSPy has built-in support for ColBERTv2, Pyserini, and Pinecone retrievers, and we have explored experimental `dspy.SQL` for executing SQL queries and `dspy.PythonInterpreter` for executing Python code in a sandbox.

---

[5]These generalize prompting techniques from the literature, respectively, by Wei et al. (2022), Chen et al. (2022), Yoran et al. (2023), and Yao et al. (2022).

[6]To highlight modularity, we use `ChainOfThought` as a drop-in replacement of the basic `Predict`. One can now simply write `RAG()("Where is Guaraní spoken?")` to use it. Notice that, if we use a signature `"context, question -> search_query"`, we get a system that generates search queries rather than answers.

```
1 class RAG(dspy.Module):
2    def __init__(self, num_passages=3):
3        # `Retrieve` will use the user's default retrieval settings unless overriden.
4        self.retrieve = dspy.Retrieve(k=num_passages)
5        # `ChainOfThought` with signature that generates answers given retrieval & question.
6        self.generate_answer = dspy.ChainOfThought("context, question -> answer")
7    def forward(self, question):
8        context = self.retrieve(question).passages
9        return self.generate_answer(context=context, question=question)
```

### 3.3 TELEPROMPTERS CAN AUTOMATE PROMPTING FOR ARBITRARY PIPELINES

When compiling a DSPy program, we generally invoke a *teleprompter*, which is an optimizer that takes the program, a training set, and a metric—and returns a new optimized program. Different teleprompters (Sec 4) apply different strategies for optimization, typically gradient-free. In DSPy, training sets may be *small*, potentially a handful of examples, though larger data enables more powerful optimization. Training examples may be *incomplete*, i.e., only *input* values are necessary. Labels for the pipeline steps are not required, unless they need to be used in the metric. In practice, we typically assume labels only for (at most) the program's final output, not the intermediate steps. This label-efficiency is critical for modularity: building a new pipeline in DSPy requires simply *recompiling* the new pipeline's code, not annotating data specific to the new pipeline.

Metrics can be simple notions like exact match (EM) or F1, or they can be entire DSPy programs that balance multiple concerns. For example, we may compile the RAG module above against a dataset of question–answer pairs `qa_trainset` and the metric EM. The goal of optimization here is to effectively bootstrap few-shot demonstrations. The following code achieves this:

```
1 # Small training set with only questions and final answers.
2 qa_trainset = [dspy.Example(question="What is the capital of France?", answer="Paris")]
3
4 # The teleprompter will bootstrap missing labels: reasoning chains and retrieval contexts.
5 teleprompter = dspy.BootstrapFewShot(metric=dspy.evaluate.answer_exact_match)
6 compiled_rag = teleprompter.compile(RAG(), trainset=qa_trainset)
```

In this example, the `BootstrapFewShot` teleprompter (Sec 4, Appendix G.2) simulates RAG on the training example(s). It will collect *demonstrations* of each module (i.e., examples of its input–output behavior) that collectively lead to valid output (i.e., respecting the signatures and the metric).

Teleprompters can be composed by specifying a `teacher` program. DSPy will sample demonstrations from this program for prompt optimization. This composition can enable very rich pipelines, where expensive programs (e.g., ensembles of large LMs) supervise cheap programs (e.g., simpler pipelines using smaller LMs). One may start with `compiled_rag` from above (say, compiled to use a large Llama2-13b-chat LM) but now fine-tune Flan-T5-large to create an efficient program:

```
1 # Larger set of questions with *no labels*. Labels for all steps will be bootstrapped.
2 unlabeled_questions = [dspy.Example(question="What is the capital of Germany?"), ...]
3 # As we assume no answer, we use `answer_passage_match` to filter ungrounded answers.
4 finetuning_teleprompter = BootstrapFinetune(metric=dspy.evaluate.answer_passage_match)
5 compiled_rag_via_finetune = finetuning_teleprompter.compile(RAG(), teacher=compiled_rag,
      trainset=unlabeled_questions, target='google/flan-t5-large')
```

## 4 THE DSPY COMPILER

A key source of DSPy's expressive power is its ability to compile—or automatically optimize—any program. Compiling relies on a teleprompter, which is an optimizer for DSPy programs that improves the quality (or cost) of modules via prompting or finetuning, which are unified in DSPy. The compiler first finds all unique `Predict` modules (predictors) in a program, including those nested under other modules. While DSPy does not enforce this when creating new teleprompters, typical teleprompters go through three stages.

**Stage 1: Candidate Generation** For *each* predictor $p$, the teleprompter may generate candidate values for the parameters of $p$: the instructions, field descriptions, or demonstrations (i.e., example input–output pairs). In this iteration of DSPy, we focus on demonstrations and find that simple rejection sampling can help bootstrap highly effective multi-stage systems. Consider the simplest non-trivial teleprompter in DSPy, `BootstrapFewShot` (simplified pseudocode in Appendix H.1).

This teleprompter will simulate a teacher program (or, if unset, the zero-shot version of the program) on some training inputs, possibly one or more times with a high temperature. When running in `compile` mode, multi-stage traces are tracked transparently and in a thread-safe fashion throughout execution. The program's metric is used to filter for multi-stage traces that together help the pipeline pass the metric. We thus obtain potential labels for all signatures in the program by throwing away the bad examples and using the good examples as potential demonstrations, though these design decisions are under user control. While LMs can be highly unreliable, we find they can be rather efficient at searching the space of solutions for multi-stage designs. A well-decomposed program can typically find at least a few training examples where the LM can pass the constraints enforced by the signatures and metrics, allowing us to bootstrap iteratively if needed.

**Stage 2: Parameter Optimization** Now each parameter has a discrete set of candidates: demonstrations, instructions, etc. Many hyperparameter tuning algorithms (e.g., random search or Tree-structured Parzen Estimators as in HyperOpt (Bergstra et al., 2013) and Optuna (Akiba et al., 2019)) can be applied for selection among candidates. We report a simplified implementation of DSPy's `BootstrapFewShotWithRandomSearch` in Appendix H.2. Another type of optimization is *finetuning* with `BootstrapFinetune`, where the demonstrations are used to update the LM's weights for each predictor. When this is applied, the LM of each module is updated to the new LM weights.

**Stage 3: Higher-Order Program Optimization** A different type of optimization that the DSPy compiler supports is modifying the control flow of the program. One of the simplest forms of this is ensembling, which we use in the case studies in this work. An ensemble will bootstrap multiple copies of the same program, and then replace the program with a new one that runs them all in parallel and *reduces* their predictions into one with a custom function (e.g., majority voting). In future work, this stage can easily accommodate techniques for more dynamic (i.e., test-time) bootstrapping as well as automatic backtracking-like logic.

## 5 GOALS OF EVALUATION

Programming frameworks can be evaluated along many dimensions: computational efficiency, developer efficiency, intuitiveness of the code and concepts, and so forth. In this paper, we focus on perhaps the most pressing issue for current LM pipelines: the role of hand-written, task-specific prompts in achieving performant systems. Our evaluations seek to test the following hypotheses:

**H1** With DSPy, we can replace hand-crafted prompt strings with concise and well-defined modules, without reducing quality or expressive power.

**H2** Parameterizing the modules and treating prompting as an optimization problem makes DSPy better at adapting to different LMs, and it may outperform expert-written prompts.

**H3** The resulting modularity makes it possible to more thoroughly explore complex pipelines that have useful performance characteristics or that fit nuanced metrics.

Our evaluation will explore these hypotheses using diverse task–program pairs. We hope this begins a shift from underspecified questions like "how do different LMs compare on GSM8K" toward "how they compare on GSM8K with program P when compiled with strategy S", which is a well-defined and reproducible run. Ultimately, our goal is to reduce the role of artful prompt construction in modern AI, in favor of the development of new modular, composable programs and optimizers.

## 6 CASE STUDY: MATH WORD PROBLEMS

We evaluate on the popular GSM8K dataset with grade school math questions (Cobbe et al., 2021). We sample 200 and 300 question–answer pairs from the official training set for training and development, respectively. Our final evaluations use the 1.3k official test set examples. We report extensive comparisons on the development set to avoid overfitting on test. Following prior work on GSM8K, we evaluate the accuracy of the final numerical value that appears in the LM output.

**Programs Considered** For this task, we consider three simple DSPy programs: a one-step Predict module (vanilla), a two-step ChainOfThought module (CoT), and finally a multi-stage ComparerOfThoughts module (ThoughtReflection). These are defined by the code below. In `reflection`, five reasoning chains are sampled from the LM (alongside their answers) and they are compared in parallel by a built-in `MultiChainComparison` module, which generalizes Yoran

Table 1: Results with in-context learning on GSM8K math word problems. Each row represents a separate pipeline: the module in the Program column is compiled against the examples in the Training set. The programs, compilers, and (small) training sets are defined in Section 6. Rows with ensemble build on the immediately preceding row. Notably, all programs in this table are expressed by composing two to four DSPy modules and teleprompters. Compiling the right *modules*, instead of string prompts, improves different LMs from 9–25% accuracy to 46–81% accuracy.

| | | | GPT-3.5 | | Llama2-13b-chat | |
|---|---|---|---|---|---|---|
| Program | Compilation | Training | Dev | Test | Dev | Test |
| vanilla | none | n/a | 24.0 | 25.2 | 7.0 | 9.4 |
| | fewshot | trainset | 33.1 | – | 4.3 | – |
| | bootstrap | trainset | 44.0 | – | 28.0 | – |
| | bootstrap×2 | trainset | 64.7 | 61.7 | 37.3 | 36.5 |
| | +ensemble | trainset | 62.7 | 61.9 | 39.0 | 34.6 |
| CoT | none | n/a | 56.0 | – | 26.7 | – |
| | fewshot | trainset | 65.1 | – | 27.3 | – |
| | fewshot | +human_CoT | 78.6 | 72.4 | 34.3 | 33.7 |
| | bootstrap | trainset | 80.3 | 72.9 | 43.3 | – |
| | +ensemble | trainset | **88.3** | **81.6** | 43.7 | – |
| reflection | none | n/a | 65.0 | – | 36.7 | – |
| | fewshot | trainset | 71.7 | – | 36.3 | – |
| | bootstrap | trainset | 83.0 | 76.0 | 44.3 | 40.2 |
| | +ensemble | trainset | 86.7 | – | **49.0** | **46.9** |

et al. (2023). This generates a new answer taking into account the patterns from the five attempts. Critically, the modules used are all generic, none is specific to math problems or a particular LM.

```
1 vanilla = dspy.Predict("question -> answer")  # GSM8K Program 'vanilla'
2 CoT = dspy.ChainOfThought("question -> answer")  # GSM8K Program 'CoT'
3
4 class ThoughtReflection(dspy.Module):
5     def __init__(self, num_attempts):
6         self.predict = dspy.ChainOfThought("question -> answer", n=num_attempts)
7         self.compare = dspy.MultiChainComparison('question -> answer', M=num_attempts)
8
9     def forward(self, question):
10        completions = self.predict(question=question).completions
11        return self.compare(question=question, completions=completions)
12
13 reflection = ThoughtReflection(num_attempts=5) # GSM8K Program 'reflection'
```

**Compiling** As we discussed in Section 4, DSPy programs can be compiled into new, optimized programs. In our experiments, we evaluate the programs zero-shot (no compiling) as well as a number of strategies for compiling. Our simplest compiler is LabeledFewShot:

```
1 fewshot = dspy.LabeledFewShot(k=8).compile(program, trainset=trainset)
```

Here, program can be any DSPy module. This samples k=8 demonstrations from the trainset for the fields common to the training examples and the signature(s), in this case, question and answer but not reasoning for instance. We report the average of 3–5 runs (depending on the setting) when applying such sampling. Next, we consider bootstrapping few-shot examples with random search:

```
1 tp = BootstrapFewShotWithRandomSearch(metric=gsm8k_accuracy)
2 bootstrap = tp.compile(program, trainset=trainset, valset=devset)
```

This will generate demonstration chains for examples in the training set and optimize the selection of demonstrations (from this set) to self-improve the program's modules. As the name indicates, this is done with random search, treating the selection of demonstrations as a parameter to optimize. Next, if desired, this bootstrapping process can be nested in DSPy. In particular, we can use the optimized bootstrap program itself to further bootstrap another program. This is relevant, for example, whenever the original zero-shot program performs relatively poorly.

```
1 bootstrap2 = tp.compile(program, teacher=bootstrap, trainset=trainset, valset=devset)
```

And lastly, we consider *ensembling* these bootstraps:

```
1 # A program that ensembles the top-7 candidate programs from a bootstrapping compiler run
      (in particular 'bootstrap' or, when applicable, 'bootstrap2') with majority voting.
2 ensemble = Ensemble(reduce_fn=dspy.majority).compile(bootstrap.programs[:7])
```

GSM8K includes human reasoning chains. Above, `trainset` does not include these reasoning chains. We also evaluate with `trainset_human_CoT`, which extends the examples in `trainset` with the human reasoning string. These two datasets can be used interchangeably as the value for the `trainset` parameter above. We note here that compiling generally runs on the order of minutes (or tens of minutes) as even the more expensive settings only require running the program a few thousand times (e.g., 10–20 trials over 150–300 validation examples) and they can occur in parallel.

**Results** Our results are summarized in Table 1, which includes dev results as well as our evaluation of promising representatives of each approach on the test set. First, the `reflection` program while only a few lines longer than the others is a clear winner, though `CoT` is quite effective with `ensemble`. Second, the `bootstrap` compilation procedure leads to large gains for every program, across both LMs. Interestingly, `vanilla` is helped by compiling with `bootstrap` as the teacher program (`bootstrap×2`). On inspecting the prompts bootstrapped (Appendix I), we see that the LM leverages the metric (i.e., correctness of the final numerical value in the output) so it uses the answer field for reasoning first. Third, while the human reasoning chains (+human_CoT) provide a large boost for `fewshot`, we can match or surpass this using `bootstrap`, which substantiates our hypothesis that DSPy can cut the need for hand-crafted prompts.[7]

## 7 CASE STUDY: COMPLEX QUESTION ANSWERING

In this case study, we explore the multi-hop question answering task with the HotPotQA (Yang et al., 2018) dataset in the open-domain "fullwiki" setting. For retrieval, we use a search index of the official Wikipedia 2017 "abstracts" dump of HotPotQA. Search is conducted by a ColBERTv2 (Santhanam et al., 2021) retriever. The HotPotQA test set is hidden, so we reserve the official validation set for our testing, and sample 1000 examples for that. We sub-divide the training set into 70%/30% train/validation splits. In the training (and thus validation) split, we keep only examples marked as "hard" in the original dataset, which matches the designation of the official validation and test sets. For training and for reporting development results, we sample 200 and 300 examples respectively.

**Programs Considered** Our simplest baseline is the `vanilla` program used in the previous case study on GSM8K (Section 6); the `"question -> answer"` signature is universal enough that it will work for many tasks when compiled appropriately. Our baseline RAG program is the one given in Section 3.2 with a `dspy.ChainOfThought` layer. This program does not excel at HotPotQA, and this motivates us to evaluate two multi-hop programs. We first test ReAct (Yao et al., 2022), a multi-step agent for tool use, which is implemented as a built-in module in DSPy. In the simplest case, a ReAct module for a particular signature can be declared as follows in DSPy:

```
react = dspy.ReAct("question -> answer", tools=[dspy.Retrieve(k=1)], max_steps=5)
```

And we test the following custom program, which is akin to Baleen (Khattab et al., 2021a), IRRR (Qi et al., 2020), and has similarities to IRCoT (Trivedi et al., 2022):

```
class BasicMultiHop(dspy.Module):
    def __init__(self, passages_per_hop):
        self.retrieve = dspy.Retrieve(k=passages_per_hop)
        self.generate_query = dspy.ChainOfThought("context, question -> search_query")
        self.generate_answer = dspy.ChainOfThought("context, question -> answer")

    def forward(self, question):
        context = []

        for hop in range(2):
            query = self.generate_query(context=context, question=question).search_query
            context += self.retrieve(query).passages

        return self.generate_answer(context=context, question=question)

multihop = BasicMultiHop(passages_per_hop=3)
```

---

[7]We can informally compare these with published results for GSM8K. Zhang et al. (2022) reports 48% for text-davinci-002, close to our results using llama2-13b, and 63% with codex using CoT. Zhang et al. (2022) report 57% for CoT prompting with PaLM 540-B, which becomes 74% upon adding self-consistency. Llama2 (Touvron et al., 2023) presented 28.7%, 42.2%, and 56.8% for llama2-13b, llama2-34b, and llama2-70b. Our implementation with llama2-13b is competitive with their llama2-34b results, while we don't use human reasoning chains in our program. Zhao et al. (2023b) reports that CoT scores around 80% for gpt-3.5-turbo from April 2023. The GPT-4 paper (OpenAI, 2023) reports that GPT-3.5 scores 57% and GPT-4 elevates this to 92% but they note that GPT-4 was trained on a subset of GSM8K.

Table 2: Results with in-context learning on HotPotQA. We report answer exact match (Ans) and pair-retrieval accuracy (Psg). Each row represents a separate pipeline: the module in the Program column is compiled against the examples in the Training set. The programs, compilers, and (small) training sets are defined in the main text. For HotPotQA, we use the training set (and not dev) directly for cross-validation. *The marked result is evaluated on 50% of our test set due to cost.

| Program | Compiler | GPT-3.5 | | | | Llama2-13b-chat | | | |
| | | Dev | | Test | | Dev | | Test | |
| | | Ans | Psg | Ans | Psg | Ans | Psg | Ans | Psg |
|---|---|---|---|---|---|---|---|---|---|
| vanilla | fewshot | 34.3 | n/a | 31.5 | n/a | 27.5 | n/a | 21.8 | n/a |
| CoT_RAG | fewshot | 36.4 | 36.0 | 29.8 | 34.4 | 34.5 | 36.0 | 28.0 | 34.4 |
| | bootstrap | 42.3 | 36.0 | – | – | 38.3 | 36.0 | 32.9 | 34.4 |
| react | none | 20.3 | – | – | – | 20.0 | – | – | – |
| | +human_r | 33.0 | – | – | – | 28.3 | – | – | – |
| | bootstrap | 31.0 | – | – | – | 24.7 | – | – | – |
| | bootstrap×2 | 39.0 | – | – | – | 40.0 | – | – | – |
| multihop | fewshot | 36.9 | 38.3 | 31.2 | 40.8 | 34.7 | 32.0 | 31.3 | 30.8 |
| | bootstrap | **48.7** | **47.0** | **39.6** | **43.8** | **42.0** | **48.3** | **36.4** | **43.5** |
| | ensemble | **54.7** | – | **45.6*** | – | **50.0** | – | **41.0** | – |

**Compiling** We continue to use the compilers we used for GSM8K (see Section 6). We also consider two compositions of our teleprompters. For ReAct, we consider applying `BootstrapFewShotWithRandomSearch` starting from an earlier bootstrap of the ReAct program. For the simple `multihop` program, we consider fine-tuning starting from its earlier bootstrap.

```
multihop_t5 = dspy.BootstrapFinetune(metric=answer_exact_match).compile(program,
    teacher=bootstrap, trainset=trainset, target='t5-large')
```

**Results** Table 2 summarizes our results: a simple `multihop` program is the best across all models and in general `bootstrap` again proves to be very effective. Perhaps most importantly, we can make Llama2-13b-chat competitive with GPT-3.5 using the `multihop` program. We also evaluated the compiler `multihop_t5` defined above which produces a T5-Large (770M parameter) model. This program scores 39.3% answer EM and 46% passage accuracy on the dev set, using only 200 labeled inputs and 800 unlabeled questions otherwise, with a teacher program consisting of an ensemble (union) of two Llama2-13b-chat multihop programs. Such a program would impose orders of magnitude lower costs for inference than a proprietary LM like GPT-3.5.[8]

## 8  CONCLUSION

This paper introduced DSPy, a new programming model for designing AI systems using pipelines of pretrained LMs and other tools. We presented three new concepts introduced in this abstraction (DSPy signatures, modules, and teleprompters), and showed in two very different case studies that it supports rapid development of highly effective systems that use relatively small LMs. We have maintained open-source versions of this framework throughout 2023 and since. In this period, we have seen and created a large number of programs that were compiled to high-quality systems by DSPy, spanning tasks from information extraction to low-resource synthetic data generation. In the interest of space and to maintain reasonable scope in this paper, we leave reporting on such tasks under controlled experimental conditions to future work. While in-context learning has proved transformative, we argue that the true expressive power in this emerging paradigm is in building sophisticated text transformation graphs in which composable modules and optimizers (teleprompters) come together to leverage LMs in more systematic and reliable ways.

---

[8]Our results may be pegged against the evaluation on HotPotQA in a number of recent papers, though there can be significant variation in evaluation methodology and test set samples across studies in this space. Si et al. (2022) achieve 25.2% EM with CoT prompting. With a "recite-and-answer" technique for PaLM-62B (Chowdhery et al., 2022), Sun et al. (2022) achieve 26.5% EM. Wang et al. (2022a) achieve 33.8% EM and 44.6 F1 when applying self-consistency for PaLM-540B. Yao et al. (2022) achieve 35.1% EM using the ReAct agent, with a tool giving it the ability for search using a Wikipedia API. Trivedi et al. (2022) reports 49% using a pipeline with code-davinci-002 on a sample of 500 HotPotQA questions.

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

# A  LATENCY & THROUGHPUT

In the main paper, we introduced the DSPy programming model and presented quality-oriented case studies. In this appendix, we report on efficiency-oriented metrics. We select one of the most sophisticated programs we discussed, namely, the `multihop` program from Sec 7.

We run a self-contained experiment from scratch to sidestep the effect of caching. We compile the `multihop` program using the latest OpenAI GPT-3.5 model at the time of writing this appendix, i.e. `gpt-3.5-turbo-1106`. We compile using the `BootstrapFewShotWithRandomSearch` teleprompter with 7 candidate programs. DSPy supports parallel evaluation. In these runs, we set the maximum number of parallel threads to 10.

Compiling this program takes approximately 370 seconds (or **6 minutes**), which is a little shorter than one minute per candidate program. The number of candidates for random search is a simple configurable parameter, whose typical range is 5 through 30. For OpenAI models, our main experiments use 13 candidate, though we observe strong results with as little as 7. This compilation process makes approximately 3200 API calls to the language model, with a total of 2.7M input tokens and 156k output tokens. With the current pricing of OpenAI models, this costs a total of **$3.0 USD** to compile.

For inference, single-threaded inference with the `multihop` program takes on the order of 2–3 seconds per question, which includes three chain of thought prompts (i.e., two for query generation and one for answer generation) and two retrieval queries (i.e., each retrieving three passages). In the presence of multiple questions, multi-threading is again possible, and we observe that we can process more than **150 questions per minute** with 10 threads. The exact maximum depends on the rate limits supported by the retriever and the LM server.

To provide a comparison of inference costs, we took four rows of increased complexity for Hot-PotQA (Table 2) and tested them with gpt-3.5-1106, while disabling the cache (which would otherwise skip any repeated computations, e.g. retrieval queries or LM calls). We ran 100 questions from HotPotQA with a single thread, and we report the average latency below. Each numbered bullet below represents a program (with compiler strategy in parentheses). For Vanilla (fewshot), we observe average latency of 0.3 seconds per question and average Cost of $0.0005 per question (1x). For CoT_RAG (fewshot), we observe average latency of 1.1 seconds per question and average cost of $0.0013 per question (2.6x). For Multihop (fewshot), we observe average latency of 2.6 seconds per question and average cost of $0.0018 per question (3.6x). For Multihop (bootstrap), we observe average latency of 2.6 seconds per question and average cost of $0.0041 per question (8.2x). As this shows, these programs are within an order of magnitude of the cost and latency of the simplest one, even though we see major quality improvements from vanilla to multihop.

Latency and throughput results for `gpt-3.5-turbo-1106` depend on the OpenAI API load and the rate limit permitted. For local models at the scale of 13B parameters, we typically see higher latencies and lower throughputs in practice, when serving the model from one or a few A100 GPUs. We suspect this gap will be bridged with recent open research advancements in serving open models, especially approaches that apply intelligent prefix caching for prompts that share the same prefixes, e.g. Zheng et al. (2023).

# B  ADVANCED SIGNATURES

When more control is desired, one can express signatures as Python classes to provide explicit instructions of the transformation and describe the format or role of each field more directly. For instance, the following signature generates search queries using context and an optional question:

```python
class GenerateSearchQuery(dspy.Signature):
    """Write a simple search query that will help answer a complex question."""

    context = dspy.InputField(desc="may contain relevant facts")
    question = dspy.InputField()
    query = dspy.OutputField(prefix="Search Query:")
```

Using the above, we can specify a complete system for the generation of a synthetic IR dataset where the queries are mediated by a question generated by the LM:

```
1 query_gen = dspy.Predict(GenerateSearchQuery)
2 query_gen(context="Language typology")
3 # Out: Prediction(question='What are the main types of language classification?',
       query='"language classification" OR "language typology" -wikipedia')
```

If questions are available, they can be supplied as shown: query_gen(context="Language typology", question="What are the primary language families of South America?").

## C  MORE ADVANCED METRICS

If one wanted to push the compiled program to be extractive given its retrieved contexts, one could define a custom metric to use in place of dspy.evaluate.answer_exact_match:

```
1 def my_rag_validation_logic(example, pred, trace=None):
2     answer_match = dspy.evaluate.answer_exact_match(example, pred)
3     # Is the prediction a substring of some passage?
4     context_match = any((pred.answer.lower() in c) for c in pred.context)
5     return answer_match and context_match
```

## D  COMPARISON WITH EXISTING LIBRARIES LIKE LANGCHAIN AND LLAMAINDEX

LangChain and LlamaIndex are perhaps the most popular library in the general space of prompting LMs. These libraries have a different focus compared to DSPy and they suffer internally from the prompt engineering challenges that DSPy aims to resolve. In particular, whereas the goal of DSPy is to tackle the fundamental challenges of prompt engineering for building new LM computational graphs, LangChain and LlamaIndex generally help application developers who need pre-packaged components and chains, e.g., implementations of popular and reusable pipelines (e.g., particular agents and specific retrieval pipelines) and tools (e.g., connections to various databases and implementations of long- and short-term memory for agents).

These off-the-shelf higher-level abstractions contrast with DSPy's focus on introducing core composable operators. In particular, DSPy introduces signatures (to abstract prompts), modules (to abstract prompting techniques), and teleprompters to act as optimizers for arbitrary imperative code (DSPy programs) that chain modules together. Its goal is to help researchers and practitioners build new LM pipelines quickly and achieve very high quality through automatic compilation (self-improvement) instead of manual prompt engineering.

In contrast, typical existing research implementations and existing libraries like LangChain and LlamaIndex are implemented using manual prompt engineering, which is the key problem that DSPy tackles. We conducted an informal study to highlight this. In late September 2023, we found that the LangChain codebase contains 50 strings exceeding 1000 characters, which are generally prompts, compared to none at all in DSPy. Indeed, a substantial number of LangChain's Python files are singularly dedicated to task-related templating and prompt engineering with 12 prompts.py files and and 42 prompt.py files. DSPy, on the other hand, provides a structured framework that automatically bootstraps prompts. The library itself does not contain a single hand-written prompt demonstration for any tasks at the time of writing, despite the very high quality with various LMs.

To review the typical forms of prompt engineering in existing libraries, we consider the following in LangChain. The LangChain Program-Aided Language Model Gao et al. (2023a) chain program uses few-shot learning, leveraging a template that is 3982 characters long with 8 math word problems (Prompt 2) and corresponding outputted programs as learning examples for the language model. LangChain also contains a prompt for SQL query tasks for *each* of the databases like Oracle, GoogleSQL, DuckDB, Crate, and MySQL, with the average length of these prompts at 1058 characters. Other task areas such as QA with sources (Prompt D) and Graph_QA also have significantly lengthy prompt templates, with averages of 1337 and 722 characters, respectively. While expert-written prompts can be useful, we believe that LM- and task-adaptive prompts bootstrapped automatically can offer far more power (and are far more modular) than hard-coding a prompt per database provider inside the code base. The next appendix section contains a number of prompts copied from related research papers and existing libraries.

## E    SAMPLE LARGE PROMPTS

This section highlights a few popular existing frameworks that structure prompts with extensive prompt engineering templates. The primary objective is to capture how many words and characters are used for such large multi-line prompts defined for tasks or tools and present these example prompts retrieved from open-sourced papers and repositories. The formatting of these example prompts is adapted from Gao et al. (2023a).

| Task/Tool Prompt | Source | Words | Characters |
|---|---|---|---|
| Prompt 1: Text-evidence checker | Gao et al. (2023a) | 818 | 4964 |
| Prompt 2: Math word problems (PAL) | LangChain & Gao et al. (2023b) | 566 | 3957 |
| Prompt 3: ReAct | Yao et al. (2022) | 593 | 3889 |
| Prompt 4: Zero-shot ReAct | LangChain | 101 | 600 |
| Prompt 5: QA with sources | LangChain | 992 | 6197 |
| Prompt 6: SQL MyScale querying | LangChain | 343 | 2239 |
| Prompt 7: Relevant docs retrieval | LlamaIndex | 129 | 719 |
| Prompt 8: IRS chatbot | LlamaIndex | 389 | 2258 |

```
1    [web] I will check some things you said.
2
3    (1) You said: Your nose switches back and forth between nostrils. When you sleep, you switch about every 45 minutes. This
     is to prevent a buildup of mucus. It's called the nasal cycle.
4    I checked: How often do your nostrils switch?
5    I found this article: Although we don't usually notice it, during the nasal cycle one nostril becomes congested and thus
     contributes less to airflow, while the other becomes decongested. On average, the congestion pattern switches about every
     2 hours, according to a small 2016 study published in the journal PLOS One.
6    Your nose's switching time is about every 2 hours, not 45 minutes.
7    This disagrees with what you said.
8
9    (2) You said: The Little House books were written by Laura Ingalls Wilder. The books were published by HarperCollins.
10   I checked: Who published the Little House books?
11   I found this article: These are the books that started it all -- the stories that captured the hearts and imaginations of
     children and young adults worldwide. Written by Laura Ingalls Wilder and published by HarperCollins, these beloved books
     remain a favorite to this day.
12   The Little House books were published by HarperCollins.
13   This agrees with what you said.
14
15   (3) You said: The Stanford Prison Experiment was conducted in the basement of Jordan Hall, Stanford's psychology building.
16   I checked: Where was Stanford Prison Experiment conducted?
17   I found this article: Carried out August 15-21, 1971 in the basement of Jordan Hall, the Stanford Prison Experiment set
     out to examine the psychological effects of authority and powerlessness in a prison environment.
18   The Stanford Prison Experiment was conducted in Jordan Hall.
19   This agrees with what you said.
20
21   (4) You said: Social work is a profession that is based in the philosophical tradition of humanism. It is an intellectual
     discipline that has its roots in the 1800s.
22   I checked: When did social work have its roots?
23   I found this article: The Emergence and Growth of the Social work Profession

 Social work's roots were planted in
     the 1880s, when charity organization societies (COS) were created to organize municipal voluntary relief associations and
     settlement houses were established.
24   Social work has its roots in the 1880s, not 1800s.
25   This disagrees with what you said.
26
27   (5) You said: The Havel-Hakimi algorithm is an algorithm for converting the adjacency matrix of a graph into its adjacency
     list. It is named after Vaclav Havel and Samih Hakimi.
28   I checked: What is the Havel-Hakimi algorithm?
29   I found this article: The Havel-Hakimi algorithm constructs a special solution if a simple graph for the given degree
     sequence exists, or proves that one cannot find a positive answer. This construction is based on a recursive algorithm.
     The algorithm was published by Havel (1955), and later by Hakimi (1962).
30   Havel-Hakimi algorithm is for constructing a special solution if a simple graph for the given degree sequence exists, or
     proving that one cannot find a positive answer, not converting the adjacency matrix of a graph into its adjacency list.
31   This disagrees with what you said.
32
33   (6) You said: "Time of My Life" is a song by American singer-songwriter Bill Medley from the soundtrack of the 1987 film
     Dirty Dancing. The song was produced by Michael Lloyd.
34   I checked: Who was the producer of "(I've Had) The Time of My Life"?
35   I found this article: On September 8, 2010, the original demo of this song, along with a remix by producer Michael Lloyd,
     was released as digital files in an effort to raise money for the Patrick Swayze Pancreas Cancer Resarch Foundation at
     Stanford University.
36   "Time of My Life" was produced by Michael Lloyd.
37   This agrees with what you said.
38
39   (7) You said: Kelvin Hopins was suspended from the Labor Party because he had allegedly sexually harassed and behaved
     inappropriately towards a Labour Party activist, Ava Etemadzadeh.
40   I checked: Why was Kelvin Hopins suspeneded from the Labor Party?
41   I found this article: A former Labour MP has left the party before an inquiry into sexual harassment allegations against
     him was able to be concluded, the party has confirmed. Kelvin Hopkins was accused in 2017 of inappropriate physical contact
     and was suspended by the Labour party pending an investigation.This agrees with what you said.
42   Kelvin Hopins was suspended because he had allegedly sexually harassed and behaved inappropriately towards a Labour Party
     activist, Ava Etemadzadeh.
43   This agrees with what you said.
44
45   (8) You said: In the battles of Lexington and Concord, the British side was led by General Thomas Smith.
46   I checked: Who led the British side in the battle of Lexington and Concord?
47   I found this article: Interesting Facts about the Battles of Lexington and Concord. The British were led by Lieutenant
     Colonel Francis Smith. There were 700 British regulars.
48   The British side was led by Lieutenant Colonel Francis Smith, not General Thomas Hall.
49   This disagrees with what you said.
50
51   (9) You said: {text}
52   I checked: {query}
53   I found this article: {evidence}
54   -----
```

Figure 1: Example few-shot prompt using a reasoning chain for agreement model that identifies inconsistencies between text and evidence (Gao et al., 2023a).

```
1   Q: Olivia has $23. She bought five bagels for $3 each. How much money does she have left?
2
3   # solution in Python:
4
5
6   def solution():
7       """Olivia has $23. She bought five bagels for $3 each. How much money does she have left?"""
8       money_initial = 23
9       bagels = 5
10      bagel_cost = 3
11      money_spent = bagels * bagel_cost
12      money_left = money_initial - money_spent
13      result = money_left
14      return result
15
16
17
18
19
20  Q: Michael had 58 golf balls. On tuesday, he lost 23 golf balls. On wednesday, he lost 2 more. How many golf balls did he
    have at the end of wednesday?
21
22  # solution in Python:
23
24
25  def solution():
26      """Michael had 58 golf balls. On tuesday, he lost 23 golf balls. On wednesday, he lost 2 more. How many golf balls
    did he have at the end of wednesday?"""
27      golf_balls_initial = 58
28      golf_balls_lost_tuesday = 23
29      golf_balls_lost_wednesday = 2
30      golf_balls_left = golf_balls_initial - golf_balls_lost_tuesday - golf_balls_lost_wednesday
31      result = golf_balls_left
32      return result
33
34
35
36
37
38  Q: There were nine computers in the server room. Five more computers were installed each day, from monday to thursday.
    How many computers are now in the server room?
39
40  # solution in Python:
41
42
43  def solution():
44      """There were nine computers in the server room. Five more computers were installed each day, from monday to thursday.
    How many computers are now in the server room?"""
45      computers_initial = 9
46      computers_per_day = 5
47      num_days = 4
48      computers_added = computers_per_day * num_days
49      computers_total = computers_initial + computers_added
50      result = computers_total
51      return result
52
53
54
55
56
57  Q: Shawn has five toys. For Christmas, he got two toys each from his mom and dad. How many toys does he have now?
58
59  # solution in Python:
60
61
62  def solution():
63      """Shawn has five toys. For Christmas, he got two toys each from his mom and dad. How many toys does he have now?"""
64      toys_initial = 5
65      mom_toys = 2
66      dad_toys = 2
67      total_received = mom_toys + dad_toys
68      total_toys = toys_initial + total_received
69      result = total_toys
70      return result
71
72
73
74
75
76  Q: Jason had 20 lollipops. He gave Denny some lollipops. Now Jason has 12 lollipops. How many lollipops did Jason give to
    Denny?
77
78  # solution in Python:
79
80
81
```

```
1
2
3
4    def solution():
5        """Jason had 20 lollipops. He gave Denny some lollipops. Now Jason has 12 lollipops. How many lollipops did Jason give
     to Denny?"""
6        jason_lollipops_initial = 20
7        jason_lollipops_after = 12
8        denny_lollipops = jason_lollipops_initial - jason_lollipops_after
9        result = denny_lollipops
10       return result
11
12
13
14
15
16   Q: Leah had 32 chocolates and her sister had 42. If they ate 35, how many pieces do they have left in total?
17
18   # solution in Python:
19
20   def solution():
21       """Leah had 32 chocolates and her sister had 42. If they ate 35, how many pieces do they have left in total?"""
22       leah_chocolates = 32
23       sister_chocolates = 42
24       total_chocolates = leah_chocolates + sister_chocolates
25       chocolates_eaten = 35
26       chocolates_left = total_chocolates - chocolates_eaten
27       result = chocolates_left
28       return result
29
30
31
32
33
34   Q: If there are 3 cars in the parking lot and 2 more cars arrive, how many cars are in the parking lot?
35
36   # solution in Python:
37
38
39   def solution():
40       """If there are 3 cars in the parking lot and 2 more cars arrive, how many cars are in the parking lot?"""
41       cars_initial = 3
42       cars_arrived = 2
43       total_cars = cars_initial + cars_arrived
44       result = total_cars
45       return result
46
47
48
49
50
51   Q: There are 15 trees in the grove. Grove workers will plant trees in the grove today. After they are done, there will be
     21 trees. How many trees did the grove workers plant today?
52
53   # solution in Python:
54
55
56   def solution():
57       """There are 15 trees in the grove. Grove workers will plant trees in the grove today. After they are done, there will
     be 21 trees. How many trees did the grove workers plant today?"""
58       trees_initial = 15
59       trees_after = 21
60       trees_added = trees_after - trees_initial
61       result = trees_added
62       return result
63
64
65
66
67
68   Q: {question}
69
70   # solution in Python:
```

Figure 2: PAL example few-shot prompt for solving math questions by generating code.

```
1   Solve a question answering task with interleaving Thought, Action, Observation steps. Thought can reason about the current
    situation, and Action can be three types:
2   (1) Search[entity], which searches the exact entity on Wikipedia and returns the first paragraph if it exists. If not, it
    will return some similar entities to search.
3   (2) Lookup[keyword], which returns the next sentence containing keyword in the current passage.
4   (3) Finish[answer], which returns the answer and finishes the task.
5   Here are some examples.
6   Question: What is the elevation range for the area that the eastern sector of the Colorado orogeny extends into?
7   Action 1: Search[Colorado orogeny]
8   Observation 1: The Colorado orogeny was an episode of mountain building (an orogeny) in Colorado and surrounding areas.
9   Action 2: Lookup[eastern sector]
10  Observation 2: (Result 1 / 1) The eastern sector extends into the High Plains and is called the Central Plains orogeny.
11  Action 3: Search[High Plains]
12  Observation 3: High Plains refers to one of two distinct land regions:
13  Action 4: Search[High Plains (United States)]
14  Observation 4: The High Plains are a subregion of the Great Plains. From east to west, the High Plains rise in elevation
    from around 1,800 to 7,000 ft (550 to 2,130 m).[3]
15  Action 5: Finish[1,800 to 7,000 ft]
16  Question: Musician and satirist Allie Goertz wrote a song about the "The Simpsons" character Milhouse, who Matt Groening
    named after who?
17  Action 1: Search[Milhouse]
18  Observation 1: Milhouse Mussolini Van Houten is a recurring character in the Fox animated television series The Simpsons
    voiced by Pamela Hayden and created by Matt Groening.
19  Action 2: Lookup[named after]
20  Observation 2: (Result 1 / 1) Milhouse was named after U.S. president Richard Nixon, whose middle name was Milhous.
21  Action 3: Finish[Richard Nixon]
22  Question: Which documentary is about Finnish rock groups, Adam Clayton Powell or The Saimaa Gesture?
23  Action 1: Search[Adam Clayton Powell]
24  Observation 1: Could not find [Adam Clayton Powell]. Similar: ['Adam Clayton Powell III', 'Seventh Avenue (Manhattan)',
    'Adam Clayton Powell Jr. State Office Building', 'Isabel Washington Powell', 'Adam Powell', 'Adam Clayton Powell (film)',
    'Giancarlo Esposito'].
25  Action 2: Search[Adam Clayton Powell (film)]
26  Observation 2: Adam Clayton Powell is a 1989 American documentary film directed by Richard Kilberg. The film is about the
    rise and fall of influential African-American politician Adam Clayton Powell Jr.[3][4] It was later aired as part of the
    PBS series The American Experience.
27  Action 3: Finish[The Saimaa Gesture]
28  Question: What profession does Nicholas Ray and Elia Kazan have in common?
29  Action 1: Search[Nicholas Ray]
30  Observation 1: Nicholas Ray (born Raymond Nicholas Kienzle Jr., August 7, 1911 { June 16, 1979) was an American film
    director, screenwriter, and actor best known for the 1955 film Rebel Without a Cause.
31  Action 2: Search[Elia Kazan]
32  Observation 2: Elia Kazan was an American film and theatre director, producer, screenwriter and actor.
33  Action 3: Finish[director, screenwriter, actor]
34  Question: Which magazine was started first Arthur's Magazine or First for Women?
35  Action 1: Search[Arthur's Magazine]
36  Observation 1: Arthur's Magazine (1844-1846) was an American literary periodical published in Philadelphia in the 19th
    century.
37  Action 2: Search[First for Women]
38  Observation 2: First for Women is a woman's magazine published by Bauer Media Group in the USA.[1] The magazine was started
    in 1989.
39  Action 3: Finish[Arthur's Magazine]
40  Question: Were Pavel Urysohn and Leonid Levin known for the same type of work?
41  Action 1: Search[Pavel Urysohn]
42  Observation 1: Pavel Samuilovich Urysohn (February 3, 1898 â August 17, 1924) was a Soviet mathematician who is best known
    for his contributions in dimension theory.
43  Action 2: Search[Leonid Levin]
44  Observation 2: Leonid Anatolievich Levin is a Soviet-American mathematician and computer scientist.
45  Action 3: Finish[yes]
```

Figure 3: ReAct example prompt for interleaving Thought, Action, Observation steps.

```
1    Answer the following questions as best you can. You have access to the following tools:
2    Search: useful for when you need to answer questions about the world
3    Use the following format:
4    Question: the input question you must answer
5    Thought: you should always think about what to do
6    Action: the action to take, should be one of [Search]
7    Action Input: the input to the action
8    Observation: the result of the action
9    ... (this Thought/Action/Action Input/Observation can repeat N times)
10   Thought: I now know the final answer
11   Final Answer: the final answer to the original input question
12   Begin!
13   Question: {question}
14   Thought:
```

Figure 4: Langchain ReAct example prompt for interleaving Thought, Action, Observation steps.

```
1   Given the following extracted parts of a long document and a question, create a final answer with references ("SOURCES").
2   If you don't know the answer, just say that you don't know. Don't try to make up an answer.
3   ALWAYS return a "SOURCES" part in your answer.
4
5   QUESTION: Which state/country's law governs the interpretation of the contract?
6   =========
7   Content: This Agreement is governed by English law and the parties submit to the exclusive jurisdiction of the English
    courts in relation to any dispute (contractual or non-contractual) concerning this Agreement save that either party may
    apply to any court for an injunction or other relief to protect its Intellectual Property Rights.
8   Source: 28-pl
9   Content: No Waiver. Failure or delay in exercising any right or remedy under this Agreement shall not constitute a waiver
    of such (or any other) right or remedy.
10  11.7 Severability. The invalidity, illegality or unenforceability of any term (or part of a term) of this Agreement shall
    not affect the continuation in force of the remainder of the term (if any) and this Agreement.
11  11.8 No Agency. Except as expressly stated otherwise, nothing in this Agreement shall create an agency, partnership or
    joint venture of any kind between the parties.
12  11.9 No Third-Party Beneficiaries.
13  Source: 30-pl
14  Content: (b) if Google believes, in good faith, that the Distributor has violated or caused Google to violate any
    Anti-Bribery Laws (as defined in Clause 8.5) or that such a violation is reasonably likely to occur,
15  Source: 4-pl
16  =========
17  FINAL ANSWER: This Agreement is governed by English law.
18  SOURCES: 28-pl
19
20  QUESTION: What did the president say about Michael Jackson?
21  =========
22  Content: Madam Speaker, Madam Vice President, our First Lady and Second Gentleman. Members of Congress and the Cabinet.
    Justices of the Supreme Court. My fellow Americans.
23  Last year COVID-19 kept us apart. This year we are finally together again.
24  Tonight, we meet as Democrats Republicans and Independents. But most importantly as Americans.
25  With a duty to one another to the American people to the Constitution.
26  And with an unwavering resolve that freedom will always triumph over tyranny.
27  Six days ago, Russia's Vladimir Putin sought to shake the foundations of the free world thinking he could make it bend to
    his menacing ways. But he badly miscalculated.
28  He thought he could roll into Ukraine and the world would roll over. Instead he met a wall of strength he never imagined.
29  He met the Ukrainian people.
30  From President Zelenskyy to every Ukrainian, their fearlessness, their courage, their determination, inspires the world.
31  Groups of citizens blocking tanks with their bodies. Everyone from students to retirees teachers turned soldiers defending
    their homeland.
32  Source: 0-pl
33  Content: And we won't stop.
34  We have lost so much to COVID-19. Time with one another. And worst of all, so much loss of life.
35  Let's use this moment to reset. Let's stop looking at COVID-19 as a partisan dividing line and see it for what it is: A
    God-awful disease.
36  Let's stop seeing each other as enemies, and start seeing each other for who we really are: Fellow Americans.
37  We can't change how divided we've been. But we can change how we move forward|on COVID-19 and other issues we must face
    together.
38  I recently visited the New York City Police Department days after the funerals of Officer Wilbert Mora and his partner,
    Officer Jason Rivera.
39  They were responding to a 9-1-1 call when a man shot and killed them with a stolen gun.
40  Officer Mora was 27 years old.
41  Officer Rivera was 22.
42  Both Dominican Americans who'd grown up on the same streets they later chose to patrol as police officers.
43  I spoke with their families and told them that we are forever in debt for their sacrifice, and we will carry on their
    mission to restore the trust and safety every community deserves.
44  Source: 24-pl
45  Content: And a proud Ukrainian people, who have known 30 years of independence, have repeatedly shown that they will not
    tolerate anyone who tries to take their country backwards.
46  To all Americans, I will be honest with you, as I've always promised. A Russian dictator, invading a foreign country, has
    costs around the world.
47  And I'm taking robust action to make sure the pain of our sanctions is targeted at Russia's economy. And I will use every
    tool at our disposal to protect American businesses and consumers.
48  Tonight, I can announce that the United States has worked with 30 other countries to release 60 Million barrels of oil
    from reserves around the world.
49  America will lead that effort, releasing 30 Million barrels from our own Strategic Petroleum Reserve. And we stand ready
    to do more if necessary, unified with our allies.
50  These steps will help blunt gas prices here at home. And I know the news about what's happening can seem alarming.
51  But I want you to know that we are going to be okay.
52  Source: 5-pl
53  Content: More support for patients and families.
54  To get there, I call on Congress to fund ARPA-H, the Advanced Research Projects Agency for Health.
55  It's based on DARPA|the Defense Department project that led to the Internet, GPS, and so much more.
56  ARPA-H will have a singular purpose|to drive breakthroughs in cancer, Alzheimer's, diabetes, and more.
```

```
1    A unity agenda for the nation.
2    We can do this.
3    My fellow Americans|tonight , we have gathered in a sacred space|the citadel of our democracy.
4    In this Capitol, generation after generation, Americans have debated great questions amid great strife, and have done
     great things.
5    We have fought for freedom, expanded liberty, defeated totalitarianism and terror.
6    And built the strongest, freest, and most prosperous nation the world has ever known.
7    Now is the hour.
8    Our moment of responsibility.
9    Our test of resolve and conscience, of history itself.
10   It is in this moment that our character is formed. Our purpose is found. Our future is forged.
11   Well I know this nation.
12   Source: 34-pl
13   =========
14   FINAL ANSWER: The president did not mention Michael Jackson.
15   SOURCES:
16
17   QUESTION: {question}
18   =========
19   {summaries}
20   =========
21   FINAL ANSWER:
```

Figure 5: Langchain example prompt for QA with sources.

```
1    You are a MyScale expert. Given an input question, first create a syntactically correct MyScale query to run, then look
     at the results of the query and return the answer to the input question.
2    MyScale queries has a vector distance function called DISTANCE(column, array) to compute relevance to the user's question
     and sort the feature array column by the relevance.
3    When the query is asking for {top_k} closest row, you have to use this distance function to calculate distance to entity's
     array on vector column and order by the distance to retrieve relevant rows.
4    *NOTICE*: DISTANCE(column, array) only accept an array column as its first argument and a NeuralArray(entity) as its second
     argument. You also need a user defined function called NeuralArray(entity) to retrieve the entity's array.
5    Unless the user specifies in the question a specific number of examples to obtain, query for at most {top_k} results using
     the LIMIT clause as per MyScale. You should only order according to the distance function.
6    Never query for all columns from a table. You must query only the columns that are needed to answer the question. Wrap
     each column name in double quotes (") to denote them as delimited identifiers.
7    Pay attention to use only the column names you can see in the tables below. Be careful to not query for columns that do
     not exist. Also, pay attention to which column is in which table.
8    Pay attention to use today() function to get the current date, if the question involves "today". ORDER BY clause should
     always be after WHERE clause. DO NOT add semicolon to the end of SQL. Pay attention to the comment in table schema.
9
10   Use the following format:
11   ======== table info ========
12   {table_info}
13   Question: {input}
14   SQLQuery:
15
16   Here are some examples:
17   ======== table info ========
18   CREATE TABLE "ChatPaper" (
19    abstract String,
20    id String,
21    vector Array(Float32),
22   ) ENGINE = ReplicatedReplacingMergeTree()
23    ORDER BY id
24    PRIMARY KEY id
25   Question: What is Feature Pyramid Network?
26   SQLQuery:   SELECT  ChatPaper.title,  ChatPaper.id,  ChatPaper.authors  FROM  ChatPaper  ORDER  BY  DISTANCE(vector,
     NeuralArray(PaperRank contribution)) LIMIT {top_k}
27
28   Let's begin:
29   ======== table info ========
30   {table_info}
31   Question: {input}
32   SQLQuery:
```

Figure 6: Langchain example prompt for SQL querying using MyScale.

```
 1   A list of documents is shown below. Each document has a number next to it along with a summary of the document. A question
     is also provided.
 2   Respond with the numbers of the documents you should consult to answer the question, in order of relevance, as well as the
     relevance score.
 3   The relevance score is a number from 1-10 based on how relevant you think the document is to the question.
 4   Do not include any documents that are not relevant to the question.
 5
 6   Example format:
 7   Document 1:
 8   <summary of document 1>
 9
10   Document 2:
11   <summary of document 2>
12
13   ...
14
15   Document 10:
16   <summary of document 10>
17
18   Question: <question>
19   Answer:
20   Doc: 9, Relevance: 7
21   Doc: 3, Relevance: 4
22   Doc: 7, Relevance: 3
23
24   Let's try this now:
25   {context_str}
26   Question: {query_str}
27   Answer:
```

Figure 7: LlamaIndex example prompt for returning relevant documents and corresponding summaries.

```
 1   You are an IRS chatbot whose primary goal is to help users with filing their tax returns for the 2022 year.
 2   Provide concise replies that are polite and professional.
 3   Answer questions truthfully based on official government information, with consideration to context provided below on
     changes for 2022 that can affect tax refund.
 4   Do not answer questions that are not related to United States tax procedures and respond with "I can only help with any
     tax-related questions you may have.".
 5   If you do not know the answer to a question, respond by saying \I do not know the answer to your question. You may be able
     to find your answer at www.irs.gov/faqs"
 6
 7   Changes for 2022 that can affect tax refund:
 8   Changes in the number of dependents, employment or self-employment income and divorce, among other factors, may affect
     your tax-filing status and refund. No additional stimulus payments. Unlike 2020 and 2021, there were no new stimulus
     payments for 2022 so taxpayers should not expect to get an additional payment.
 9   Some tax credits return to 2019 levels. This means that taxpayers will likely receive a significantly smaller refund
     compared with the previous tax year. Changes include amounts for the Child Tax Credit (CTC), the Earned Income Tax Credit
     (EITC) and the Child and Dependent Care Credit will revert to pre-COVID levels.
10   For 2022, the CTC is worth $2,000 for each qualifying child. A child must be under age 17 at the end of 2022 to be a
     qualifying child. For the EITC, eligible taxpayers with no children will get $560 for the 2022 tax year. The Child and
     Dependent Care Credit returns to a maximum of $2,100 in 2022.
11   No above-the-line charitable deductions. During COVID, taxpayers were able to take up to a $600 charitable donation tax
     deduction on their tax returns. However, for tax year 2022, taxpayers who don't itemize and who take the standard deduction,
     won't be able to deduct their charitable contributions.
12   More people may be eligible for the Premium Tax Credit. For tax year 2022, taxpayers may qualify for temporarily expanded
     eligibility for the premium tax credit.
13   Eligibility rules changed to claim a tax credit for clean vehicles. Review the changes under the Inflation Reduction Act
     of 2022 to qualify for a Clean Vehicle Credit.
```

Figure 8: LlamaIndex example prompt for IRS chatbot guidelines.

## F  SOFTWARE DEPENDENCIES

DSPy has the following core software requirements:

- **Python:** Version 3.9 or higher.
- **HuggingFace Text Generation Inference server:** Used for hosting language models from the HuggingFace transformers library.
- **OpenAI connectivity:** Required for accessing GPT models.
- **ColBERTv2 retrieval server:** Serves as the default search index retriever.

For optimal performance:

- Access to GPUs is recommended for hosting the language and retrieval models.
- DSPy experiments specifically relied on **NVIDIA A100-SXM GPUs** with 80 GiBs memory.

For model fine-tuning, the following package versions were used:

- `datasets-2.14.5`
- `transformers-4.32.0`
- `peft-0.5.0`
- `trl-0.7.1`

## G  MODULES

### G.1  PREDICT

```python
class Predict(dspy.Module):
    def __init__(self, signature, **config):
        self.signature = dspy.Signature(signature)
        self.config = config

        # Module Parameters.
        self.lm = dspy.ParameterLM(None) # use the default LM
        self.demonstrations = dspy.ParameterDemonstrations([])

    def forward(self, **kwargs):
        lm = get_the_right_lm(self.lm, kwargs)
        signature = get_the_right_signature(self.signature, kwargs)
        demonstrations = get_the_right_demonstrations(self.demonstrations, kwargs)

        prompt = signature(demos=self.demos, **kwargs)
        completions = lm.generate(prompt, **self.config)
        prediction = Prediction.from_completions(completions, signature=signature)

        if dsp.settings.compiling is not None:
            trace = dict(predictor=self, inputs=kwargs, outputs=prediction)
            dspy.settings.traces.append(trace)

        return prediction
```

### G.2  CHAIN OF THOUGHT

```python
class ChainOfThought(dspy.Module):
    def __init__(self, signature):

        # Modify signature from '*inputs -> *outputs' to '*inputs -> rationale, *outputs'.
        rationale_field = dspy.OutputField(prefix="Reasoning: Let's think step by step.")
        signature = dspy.Signature(signature).prepend_output_field(rationale_field)

        # Declare a sub-module with the modified signature.
        self.predict = dspy.Predict(self.signature)

    def forward(self, **kwargs):
        # Just forward the inputs to the sub-module.
        return self.predict(**kwargs)
```

# H  TELEPROMPTERS

## H.1  BOOTSTRAPFEWSHOT

```python
class SimplifiedBootstrapFewShot(Teleprompter):
    def __init__(self, metric=None):
        self.metric = metric

    def compile(self, student, trainset, teacher=None):
        teacher = teacher if teacher is not None else student
        compiled_program = student.deepcopy()

        # Step 1. Prepare mappings between student and teacher Predict modules.
        # Note: other modules will rely on Predict internally.
        assert student_and_teacher_have_compatible_predict_modules(student, teacher)
        name2predictor, predictor2name = map_predictors_recursively(student, teacher)

        # Step 2. Bootstrap traces for each Predict module.
        # We'll loop over the training set. We'll try each example once for simplicity.
        for example in trainset:
            if we_found_enough_bootstrapped_demos(): break

            # turn on compiling mode which will allow us to keep track of the traces
            with dspy.setting.context(compiling=True):
                # run the teacher program on the example, and get its final prediction
                # note that compiling=True may affect the internal behavior here
                prediction = teacher(**example.inputs())

                # get the trace of the all interal Predict calls from teacher program
                predicted_traces = dspy.settings.trace

            # if the prediction is valid, add the example to the traces
            if self.metric(example, prediction, predicted_traces):
                for predictor, inputs, outputs in predicted_traces:
                    d = dspy.Example(automated=True, **inputs, **outputs)
                    predictor_name = self.predictor2name[id(predictor)]
                    compiled_program[predictor_name].demonstrations.append(d)

        return compiled_program
```

## H.2  BOOTSTRAPFEWSHOTWITHRANDOMSEARCH

```python
class SimplifiedBootstrapFewShotWithRandomSearch(Teleprompter):
    def __init__(self, metric = None, trials=16):
        self.metric = metric
        self.trials = trials

    def compile(self, student, *, teacher=None, trainset, valset=None):
        # we can do forms of cross-validation if valset is unset.
        valset = trainset if valset is None else valset

        candidates = []
        for seed in range(self.trials):
            # Create a new basic bootstrap few-shot program.
            shuffled_trainset = shuffle(trainset, seed=seed)
            tp = BootstrapFewShot(metric=metric, max_bootstrap_demos=random_size())
            candidate_program = tp.compile(student, shuffled_trainset, teacher)

            # Step 2: Evaluate the generated candidate program.
            score = evaluate_program(candidate_program, self.metric, valset)
            candidates.append((score, candidate_program))

        # return the best candidate program.
        return max(candidates, key=lambda x: x[0])[1]
```

# I   EXAMPLES OF THE PROMPTS AUTOMATICALLY GENERATED BY DSPY

We include three example prompts bootstrapped by DSPy for the `llama2-13b-chat` experiments in this paper in Figures 9, 10, and 11. These include the prompt automatically generated by DSPy for GSM8K vanilla program compiled with bootstrap×2, the GSM8K CoT program compiled with bootstrap, and the second-hop `generate_query` prompt for HotPotQA's multihop program. All of these, particularly their demonstrations' labels and their selection, are generated by DSPy automatically using `llama2-13b-chat`. Consider the first one, i.e. Figures 9. This shows that, even though the program (vanilla) does not induce a chain of thought, the LM effectively leverages the nature of the GSM8K metric (checking for the final number for evaluation) and bootstraps its own reasoning-based demonstration, which makes the program effective at test time. Next, consider the second-hop query prompt bootstrapped by DSPy using `llama2-13b-chat`, shown in Figure 11. We hypothesize this prompt works effectively because it demonstrates the process of generating search queries for two complex questions by producing two very different types of queries: a simplified question ("When was the first...?") and a keyword query ("Jeremy Paxman birth year"). This might allow the LM to resolve different questions at test-time more effectively.

```
1    Given the fields 'question', produce the fields 'answer'.
2
3    ---
4
5    Follow the following format.
6
7    Question: ${question}
8    Answer: ${answer}
9
10   ---
11
12   Question: Jimmy and Irene go shopping for clothes on a Tuesday, where senior citizens get a 10% discount on their purchases.
     Jimmy picks out 3 shorts from the $15 rack. Irene grabs 5 shirts from the $17 rack. How much money do they give to the
     cashier?
13   Answer: Jimmy picks out 3 shorts at $15 each = $45. Irene grabs 5 shirts at $17 each = $85. Total cost = $45 + $85 = $130.
     Since senior citizens get a 10% discount, they will pay 10% of $130 = $13. So, they will give the cashier $130 - $13 =
     $117. ---
14
15   ---
16
17   Question:
```

Figure 9: Copy of the prompt automatically generated by DSPy for GSM8K Llama2-13b-chat vanilla program compiled with bootstrap×2.

```
1    Given the fields 'question', produce the fields 'answer'.
2
3    ---
4
5    Follow the following format.
6
7    Question: ${question}
8    Reasoning: Let's think step by step in order to $produce the answer. We ...
9    Answer: ${answer}
10
11   ---
12
13   Question: Mark is baking bread. He has to let it rise for 120 minutes twice. He also needs to spend 10 minutes kneading
     it and 30 minutes baking it. How many minutes does it take Mark to finish making the bread?
14   Reasoning: Let's think step by step in order to find out how many minutes it takes Mark to finish making the bread. We know
     that he needs to let it rise for 120 minutes twice, so that's 240 minutes. Then, he needs to spend 10 minutes kneading it
     and 30 minutes baking it. So, in total, it will take Mark 240 + 10 + 30 = 280 minutes to finish making the bread.
15   Answer: 280 ---
16
17   ---
18
19   Question: Ben has $2000 for his business operations costs. He orders goods from his supplier and writes them a cheque for
     $600. His debtor pays him $800 from the purchases they had made on credit. Mr. Ben then decides to do equipment maintenance
     and spends $1200 on the whole operation. How much money is Mr. Ben remaining with?
20   Reasoning: Let's think step by step in order to find out how much money Mr. Ben is remaining with. We know that he had
     $2000 to begin with, and he spent $600 on goods from his supplier, so he has $2000 - $600 = $1400 left. Then, his debtor
     paid him $800, so he has $1400 + $800 = $2200. Finally, he spent $1200 on equipment maintenance, so he has $2200 - $1200
     = $1000 left.
21   Answer: $1000 ---
22
23   ---
24
25   ... several other demonstrations here ...
26
27   ---
28
29   Question:
```

Figure 10: Shortened copy of the prompt automatically generated by DSPy for GSM8K Llama2-13b-chat CoT program compiled with bootstrap.

```
1    Given the fields 'context', 'question', produce the fields 'search_query'.
2
3    ---
4
5    Follow the following format.
6
7    Context: ${context}
8    Question: ${question}
9    Reasoning: Let's think step by step in order to ${produce the search_query}. We ...
10   Search Query: ${search_query}
11
12   ---
13
14   Context:
15   [1] Twilight (novel series) | Twilight is a series of four vampire-themed fantasy romance novels by American author
     Stephenie Meyer. ...
16   [2] Harper Connelly Mysteries | The Harper Connelly Mysteries is a series of fantasy mystery novels written by Charlaine
     Harris, and first published in 2005. ...
17   [3] The Dark Heroine | The Dark Heroine is a series of vampire-themed fantasy romance novels written by English author
     Abigail Gibbs, published by HarperCollins in 2012. ...
18
19   Question: In which year was the first of the vampire-themed fantasy romance novels for which The Twilight Saga: The
     Official Illustrated Guide serves as a spin-off encyclopedic reference book first published?
20
21   Reasoning: Let's think step by step in order to determine the year the first of the vampire-themed fantasy romance novels
     was first published. ...
22
23   Search Query: When was the first of the vampire-themed fantasy romance novels published?
24
25   ---
26
27   Context:
28   [1] The Victorians | The Victorians - Their Story In Pictures is a 2009 British documentary series which focuses on
     Victorian art and culture. ...
29   [2] The Caxtons | The Caxtons: A Family Picture is an 1849 Victorian novel by Edward Bulwer-Lytton that was popular in its
     time.
30   [3] Victorian (comics) | The Victorian is a 25-issue comic book series published by Penny-Farthing Press and starting in
     1999. ...
31
32   Question: The Victorians - Their Story In Pictures is a documentary series written by an author born in what year?
33
34   Reasoning: Let's think step by step in order to produce the search query. We know that the documentary series is about
     Victorian art and culture, and it was written and presented by Jeremy Paxman. Therefore, we need to find the year in which
     Jeremy Paxman was born.
35
36   Search Query: Jeremy Paxman birth year
37
38   ---
39
40
41   Context:
```

Figure 11: Shortened copy of the prompt automatically generated by DSPy for HotPotQA Llama2-13b-chat multi-hop program (generating second hop query) compiled with bootstrap.

## J  FUTURE WORK DIRECTIONS

As we discuss in Sec 2, this work is inspired by the role that abstractions and libraries for deep learning played in the development of the field. Our goal for DSPy is to recreate that revolution for what we call *foundation model programming*. To this end, we have started working on several follow-up directions that expand and improve DSPy.

At a high level, we see highly promising directions in: (1) the implementation of new modules (corresponding to various new prompting techniques), (2) the systematic evaluation of a larger number of advanced programs for open-ended tasks, (3) the addition of various automatic LM-based metrics for optimization, (4) the addition of methods for better debugging and inspection and (5) for more controlled generation (a la LMQL; Beurer-Kellner et al. 2023), and (6) the development a deeper understanding of the tradeoffs associated with compiling with large vs. small LMs.

Beyond these, one of the key strengths of DSPy is the modularity between programs and teleprompters (optimizers). The discussion in Sec 4 lays out several angles for building more sophisticated teleprompters, which can improve the quality of existing as well as new DSPy programs.

**Better candidate generation.** We envision a more systematic treatment of the variables to optimize, e.g. instructions, demonstrations, string prefixes (e.g., "Let's think step by step" for chain of thought), and numerical hyperparameters (e.g., the number of passages to retrieve or the number of candidate answers to generate). Future work may benefit from the way variables are defined in Optuna (Akiba et al., 2019) and in the conceptual framework of Dohan et al. (2022) to allow expressing these. Future work may also evaluate methods to generate more diverse or more challenging demonstrations, including more sophisticated logic for bootstrapping (e.g., trying multiple variants of the program) or filtering (e.g., only selecting examples that the zero-shot version of the program *cannot* solve correctly).

**Better parameter optimization.** More sophisticated programs will likely benefit from more advanced optimization strategies. For instance, the Optuna teleprompter in DSPy may allow Bayesian optimizations that exceed the quality achieved with random search. More research is required to understand the tradeoffs of cost and quality in this case. Going a step beyond this, LMs can take a more active role in optimization, beyond bootstrapping demonstrations. For instance, they can revise instructions based on categories of common failures or attempt to simulate a notion of gradients over multi-stage programs.

**Better higher-order program optimizations.** All programs in this paper were directed to a prompt (or ensemble of optimized prompts) for each module. It is possible to treat different inputs for each module more dynamically, directing them to more specialized prompts. While DSPy has an initial teleprompter that treats different inputs more dynamically, this space is very large and warrants much exploration. Beyond that, the ability to backtrack to correct mistakes and address LM feedback accordingly is another promising direction.

In all of these, the references presented in Sec 2 may serve as powerful starting points for each individual component (e.g., utilizing LM feedback), which DSPy can generalize and extend to deal with arbitrary LM pipelines of declarative modules.

