# OpenReview forum: "DSPy: Compiling Declarative Language Model Calls into State-of-the-Art Pipelines"
_ICLR.cc/2024/Conference — ICLR 2024 spotlight_

### Official Review · Reviewer_cQiR · 2023-10-31

**Soundness:** 3 good
**Presentation:** 4 excellent
**Contribution:** 3 good
**Rating:** 6
**Confidence:** 3

**Summary:**

This paper introduces DSPy, an LM pipeline framework consisting of a programming model and compiler. The programming model provides composable and declarative modules for LM instruction. These modules function similarly to function calls, allowing users to define input/output behavior using natural language signatures. The compiler is capable of generating high-quality prompts automatically or fine-tuning LMs using a general optimization strategy (teleprompter). Through evaluations on GSM8K and HotpotQA, the authors demonstrate DSPy's ability to reduce the required number of handwritten prompt templates.

**Strengths:**

1. Novel Approach. This paper introduces a novel approach to systematically build LM pipelines and compile modules into a set of prompts (or fine-tunes) optimized for specific tasks. The approach is promising, showcasing its ability to reduce the human effort required to develop prompt pipelines.
2. Well-Written. The paper is clearly written and presents the core concepts of DSPy in a straightforward manner.

**Weaknesses:**

1. Lack of details: DSPy stands out from other frameworks, like LangChain, due to its capacity for automatic prompt generation and optimization. However, the introduction of the DSPy Compiler lacks sufficient specificity. It would be beneficial to provide more comprehensive details on how DSPy generates candidate values for module parameters (instructions, field description, and example input/output) based on signatures.

**Questions:**

## Approach:
1. Can we easily implement common and advanced prompting techniques with signatures? I noticed that the signatures used in the Case Study are quite simple. When it comes to dealing with complex user intentions, do you think additional hand-written comments are necessary alongside signatures? For instance, in the ChainOfThought module, I found that the prompt "Let's think step by step" still requires handwriting within the module. How many prompts would need to be handwritten to achieve functional equivalence with a complex prompt (such as Appendix D prompt 6) using signatures?
2. How can users debug their DSPy programs effectively? In Section 4 of the paper, it is mentioned that "A well-decomposed program can typically find at least a few training examples..." I suppose that the success of optimization depends on a well-structured decomposition by the user, which implies the need for frequent modification of DSPy programs. Consequently, it raises the question of how users can determine which modules are not functioning correctly. Is it necessary for users to repeatedly rewrite the entire DSPy program?
3. How much does the optimization cost in terms of computing resources or API usage? (This question may exceed the paper's scope)
## Evaluation
1. In section 5, Hypothesis 2, the paper mentions the possibility of DSPy outperforming expert-written prompts. However, is there an evaluation comparing DSPy with handwritten prompts to support this claim?
2. In Section 5 of the paper, it states, "... ‘how they compare on GSM8K with program P when compiled with strategy S’, which is a well-defined and reproducible run.” How can the reproducibility of these results be ensured?
3. In Case Study 1 using the GSM8K dataset, GPT-3.5 with 5 few shots achieves a score of 57%, while vanilla+GPT-3.5+fewshot only scores 33.1%. What factors contribute to the decline in performance, considering the similarity between these two approaches?

---

> ### Author Response · Authors · 2023-11-22
>
> Thank you for the detailed feedback on our work. We are delighted that you found our approach novel and promising and our paper well-written.
>
>
> > More comprehensive details on how DSPy generates candidate values for module parameters (instructions, field description, and example input/output) based on signatures.
>
> Thank you for the suggestion. We now include a new appendix I (pages 28, 29, 30) which include snippets of our optimized prompts, when compiling with Llama2-13b-chat. This should help the reader concretely visualize the parts of a typical prompt constructed from a signature. In particular, it contains the instructions (which, if omitted, are inferred from the fields), the field organization, and the demonstrations.
>
> In short, the module defines a parameter space. A teleprompter can then optimize these different pieces. For instance, we have an ongoing research project that optimizes the instructions of DSPy modules automatically. In this paper, we focus on optimization with random search over demonstrations, where we automatically created traces of each module, and then select different traces as few-shot examples (or finetuning training data), even if there are no hand-written labels for these modules.
>
>
> > Can we easily implement common and advanced prompting techniques with signatures? When it comes to dealing with complex user intentions, do you think additional hand-written comments are necessary alongside signatures?
>
> A prompting technique is best abstracted as a Module in DSPy. It is indeed possible that a specific approach to prompting requires adding new modules to DSPy, which should work for any signature (i.e., any input/output patterns).
>
> Our vision is that any minor hard-coded parts of the modules (e.g., “Let’s think step by step.”) should eventually be treated as variables to optimize. Nonetheless, like in traditional DNNs, there are always hyperparameters that may be selected by hand, as long as the overall model at the end is optimized automatically.
>
> Since the submission of this work, we have built (or helped others build) a number of new sophisticated systems with DSPy. In some of them, we found that users like to include additional comments (or instructions) in the extended DSPy format that we present in Appendix B “Advanced Signatures”. To make this more robust, we have been exploring techniques to automatically optimize these prompts and to allow for automatic backtracking logic to correct errors. Considering the scope, length, and complexity of this paper, we leave exploring these additional components to future publications related to DSPy.
>
>
> > How can users debug their DSPy programs effectively? [...] it raises the question of how users can determine which modules are not functioning correctly. Is it necessary for users to repeatedly rewrite the entire DSPy program?
>
> One of the key reasons we built DSPy as a “define-by-run” abstraction (like PyTorch, or the “eager mode” of TensorFlow 2.0) is to facilitate debugging.
>
> In particular, users of DSPy can simply insert checkpoints or use print statements between different steps of their DSPy programs. For example, in the multi-hop program, we can print or log every search query generated (or even manually change these queries) in order to understand which module is responsible for different mistakes.
>
> Because it is possible to debug one module at a time, we do not anticipate any need for users to repeatedly rewrite their programs. Instead, they can focus on isolating individual types of errors and then improving the responsible module. Of course, DSPy is not a panacea and sometimes development or debugging may require more end-to-end tracing of the full program. To this end, we recently started adding support for tracing of the full program to visualize all the prompts during execution.
>
> [continued below]

---

> ### Author Response · Authors · 2023-11-22
>
> [continue...]
>
>
> > How much does the optimization cost in terms of computing resources or API usage?
>
> Thank you for emphasizing this element. To address this, we have added a new appendix A on Latency & Throughput of using DSPy. To illustrate that the cost of using DSPy is manageable, we consider a sophisticated program from our evaluation, i.e. the multi-hop search and QA program. We report on compiling it with OpenAI GPT-3.5 for $3.00 in just six minutes. Inference with this program takes approximately 3 seconds per example (single-threaded) and can easily exceed 150 questions per minute (with ten DSPy threads).
>
> Future work should explore efficiency in a much broader evaluation to characterize this emerging space of LM programs. Just like the cost of training a new model, the cost of compiling a DSPy can vary greatly depending on the complexity of the program and the optimizer. However, we hope this small-scale evaluation sheds some light on the typical cost of compiling a program like this.
>
>
> > In section 5, Hypothesis 2, the paper mentions the possibility of DSPy outperforming expert-written prompts. However, is there an evaluation comparing DSPy with handwritten prompts to support this claim?
>
>
> This is indeed an important aspect of DSPy. In our evaluations, the CoT program for GSM8K includes comparison between the “bootstrap” compilation approach and the “fewshot +human_CoT” setting (Table 1). The former generates its own optimized few-shot examples (including, in this program, the reasoning chains). The latter samples expert-written step-by-step reasoning chains, which are the typical way other papers build prompts for GSM8K (e.g., the original CoT paper by Wei et al). Another instance of this is the ReAct evaluations in Table 2, which compare handwritten prompting against our bootstrapping compilation approaches. Across these evaluations, we found that the compiled systems match or exceed the quality achieved by hand-written prompts and reasoning chains. These results can also be compared with systems from the literature (e.g., the original ReAct scores on HotPotQA), which generally use hand-written prompts (e.g., see Appendix E).
>
>
> >  In Section 5 of the paper, it states, "... ‘how they compare on GSM8K with program P when compiled with strategy S’, which is a well-defined and reproducible run.” How can the reproducibility of these results be ensured?
>
> To reproduce a specific program, DSPy supports saving a dump of the compiled program. This is akin to saving a trained model, to draw an analogy with deep neural networks. As long as the LM can produce reproducible outputs, loading the saved program can be used to replicate old runs, if the LLMs are set for greedy decoding (i.e., temperature=0.0). DSPy also supports caching (and sharing the cache for) all invocations of the LM(s), which is another way to ensure reproducibility as needed.
>
> More generally, it's also possible to re-run each setup several times (with higher temperature) and average the runs. Either way, like DNNs, DSPy offers a more self-contained setup than hand-drafting prompts (or hand-tuning classifier weights), which aids reproducibility of the key ideas in the the more general sense that the optimization process itself is automated and hence can be reapplied to new settings. In contrast, hand-tuning prompts is more of an art than a science and is hard to recreate.
>
>
> > In Case Study 1 using the GSM8K dataset, GPT-3.5 with 5 few shots achieves a score of 57%, while vanilla+GPT-3.5+fewshot only scores 33.1%. What factors contribute to the decline in performance, considering the similarity between these two approaches?
>
> Thank you for the remark. Let us clarify how the different approaches for few-shot prompting work on GSM8K.
>
> In Table 1, we report the following results with `gpt-3.5-turbo-instruct` (released in September 2023).
>
> 1. Using the `fewshot` (LabeledFewShot) teleprompter in DSPy, on the `vanilla` program.
> 2. Using the `fewshot` teleprompter on the `CoT` program.
> 3. Using the `fewshot` teleprompter on the `CoT` program, with the extended “+human CoT” training set.
>
> In the first two, we prompt the LM with few-shot examples that do *not* contain reasoning chains. That is, the demonstrations in #1 and #2 only include the final numerical answer. In the first one, we also do not request the model to engage in a step-by-step reasoning chain, unlike #2 and #3. In the third one, the few-shot examples include human-written reasoning chains.
>
> These three approaches score 33%, 65%, and 78% on our development set. We test the last one on the official test set, and it scores 72%. In this case, the most comparable run to the original 5-shot GPT-3.5 evaluation by OpenAI is the third option. As you can see, our implementation here scores more highly (72% > 57%), which may be a result of using a newer release of GPT-3.5 (September vs March 2023), using extra demonstrations (5-shot vs 8-shot), and variance across samples (we average 5 runs).

---

### Official Review · Reviewer_zPoD · 2023-10-31

**Soundness:** 3 good
**Presentation:** 3 good
**Contribution:** 3 good
**Rating:** 8
**Confidence:** 4

**Summary:**

This paper presents a programming model DSPy that allows defining parameterized LLM-based text transformations. Under this model, multi-step LLM strategies like chain-of-thought, ensembles of chains-of-thoughts, ReAct, and reflection are straightforward to implement. The parameters in these strategies can include the few-shot demonstrations and the specific prompting wording, and DSPy admits using LLM-powered search strategies for optimizing over these strategy parameters. With DSPy, the paper authors demonstrate using a large language model to optimize prompting strategies for use with comparatively smaller language models.

**Strengths:**

* The paper introduces a novel programming paradigm for defining and optimizing parameterized LLM-based text transformations.
* Though I have not used DSPy myself, my impression is that there is a focus on expressiveness and usability in the programming model, which enhances its usefulness for the ICLR community. One piece of evidence for this expressiveness is the ability showcased to write CoT, reflection, ReAct, and other prompting strategies concisely within the DSPy model. Another is the composibility e.g. of different programs with different optimization (aka "compilation") approaches.
* DSPy reduces the reliance of creating LLM-based text transformations on manual prompting. This can be seen as a strength, reducing the need for expertise in prompt writing. (It can also on some occasions be a weakness if it increases the amount of examples required e.g. from none.)

**Weaknesses:**

* My impression is that the value-add of DSPy for any of the listed strategies is somewhat small. E.g. implementing any of CoT, Reflection, or ReAct without DSPy does not require much code. The same is true, I think, for the optimization approaches / compilers. I expect this value-add grows when working with many such strategies at once, and additionally that DSPy provides organizational value both as the programs and compilers grow in complexity, and as the set of people using them grows. Being able to compose different strategies and compiler techniques is also one of the key beneficial properties of the system.
* The evaluations performed do not provide a measure of compute usage or time (both for compilation as well as inference of the compiled programs), which makes comparisons across programs and compilers less meaningful.
* There are no examples of compiled programs provided in the paper or appendices. I think an analysis of compiled programs would benefit the paper meaningfully. In particular, some unanswered questions about the compiled programs include: how do they differ from the types of prompt programs that people write by hand or using other frameworks? how do programs compiled for small LMs differ from those compiled for large LMs? are there any obvious patterns in the compiled programs? how about obvious shortcomings or irregularities, where additional hand-optimization would be easy? any evidence that the optimization techniques overfit to the validation set?

**Questions:**

# Questions and Suggestions:

Terminology: In the abstract you say DSPy programs are "compiled to" a language model. I think this wording is a bit wrong or misleading, and that it would be better to say the program is "optimized for" or (if you insist on the language of compilation) "compiled for".

The paper has a repeated metaphor with neural network abstractions and deep learning frameworks (Section 1 paragraph 3, and Section 2 paragraph 1, and Section 3.2 paragraph 4). The metaphor rests on two properties of DSPy: its programs are modular and admit optimization. However, the main feature of deep learning systems is that they admit *gradient based* optimization, which is absent from DSPy, weakening the metaphor.

Teleprompters are described as general-purpose optimization strategies, though if I understand correctly they are limited to gradient-free optimization techniques. This detail warrants mentioning.

Strong claim: The paper claims DSPy is the first programming model that translates prompting techniques into parameterized declarative modules that can be optimized. In evaluating this claim, I considered the following relevant works that I thought I would note here.
Dohan 2022 Language Model Cascades https://arxiv.org/abs/2207.10342 also casts several prompting techniques like scratchpads / chain of thought, verifiers, STaR, selection-inference under a unified framework.
Decomposed Prompting: A Modular Approach for Solving Complex Tasks https://arxiv.org/abs/2210.02406 splits tasks into subtasks that are solved a recombined to solve the overall task.
ANPL: Compiling Natural Programs with Interactive Decomposition https://arxiv.org/abs/2305.18498 admits the user performing the decomposition of tasks into subtasks.
Large Language Models as Optimizers https://arxiv.org/abs/2309.03409 like a DSPy compiler produces text-transformation using an LLM to perform prompt optimization.
Among these, Dohan 2022 is closest to the claim made in the paper.

-- In your treatment of Predict, I don't think the inclusion of the "None" implementation detail aids the discussion.

In the list of parameters (Section 3.2, paragraph 3), field descriptions are omitted.
How do you think about safety during SQL generation, since SQL queries can modify or delete tables. How do you approach using tools with side effects during compilation?
In the RAG example at the end of Section 3.2, is it complete without a definition of Retrieve provided?
How can a DSPy user introduce a new Retrieve implementation?

In my view, one of the key results is that expensive LLMs can be used during the optimization process of a prompt pipeline targeting smaller LMs. I will copy here the text from the "weaknesses section" above pertaining to this point: There are no examples of compiled programs provided in the paper or appendices. I think an analysis of compiled programs would benefit the paper meaningfully. In particular, some unanswered questions about the compiled programs include: how do they differ from the types of prompt programs that people write by hand or using other frameworks? how do programs compiled for small LMs differ from those compiled for large LMs? are there any obvious patterns in the compiled programs? how about obvious shortcomings or irregularities, where additional hand-optimization would be easy? any evidence that the optimization techniques overfit to the validation set?

Regarding how the compiled programs differ when targeting smaller LMs, a qualitative investigation with examples would be welcome.

In Table 1, reporting some measure of compute usage or time both for compilation and inference would be valuable, and would make the various strategies (both programs and compilation approaches) more readily comparable.

Note: MultiChainComparison implementation is not provided.
Note: Do the compilation approaches account for overfitting the validation set?

Another idea for further study: Can you compile a DSPy program for a small LM and then run that program using a more powerful LLM different from the one targeted during compilation in order to save on compilation costs? What sort of performance degradation does this incur?

I also include here some typographic issues I identified in the paper. These did not meaningfully hinder readability of the work.

Typo: GMS08K -> GSM8K (page 2, paragraph 4)
Typo: format -> formats (page 4, Section 3.2)
Typo: grammar: but -> or (page 5, Section 3.3)
Typo: "As we assumes" in the example at the end of Section 3.3
Typo: these -> this, ensembles -> ensembling (page 6, stage 3)
Typo: "specific math problems or particular LM." -> "specific to math problems or a particular LM." (page 7)

---

> ### Author Response · Authors · 2023-11-22
>
> Thank you for the thoughtful review and the in-depth feedback on our work. We are glad to see that you found our programming paradigm 'novel' and our 'focus on expressiveness and usability', as evidenced in the 'composability' of programs and optimizations, promising for the ICLR community.
>
>
> > My impression is that the value-add of DSPy for any of the listed strategies is somewhat small. E.g. implementing any of CoT, Reflection, or ReAct without DSPy does not require much code. [...] I expect this value-add grows when working with many such strategies at once, and additionally that DSPy provides organizational value both as the programs and compilers grow in complexity [...] Being able to compose different strategies and compiler techniques is also one of the key beneficial properties of the system.
>
> Thank you for the nuanced discussion. We agree that much of the power of DSPy comes from enabling rich compositions of modules and compilation strategies. However, we wish to emphasize that even simple programs (e.g., those that rely on a single module, like CoT or ReAct) stand to benefit from DSPy. To illustrate, it’s indeed true that expressing CoT (or, say, ReAct) does not require much code without DSPy. However, expressing them *well* for a particular task generally requires writing a good prompt, often with high-quality few-shot examples for reasoning or tool use. Adapting this prompt across LMs may also pose a challenge. While these challenges for a single, simple module are not very large (and indeed more research is needed to make general claims about them), we believe the DSPy paradigm offers a consistent way to optimize both simple and complex programs alike.
>
>
> > The evaluations performed do not provide a measure of compute usage or time (both for compilation as well as inference of the compiled programs)
>
> Thank you for emphasizing this element. To address this, we have added a new appendix A on Latency & Throughput of using DSPy. To illustrate that the cost of using DSPy is manageable, we consider a sophisticated program from our evaluation, i.e. the multi-hop search and QA program. We report on compiling it with OpenAI GPT-3.5 for $3.00 in just six minutes. Inference with this program takes approximately 3 seconds per example (single-threaded) and can easily exceed 150 questions per minute (with ten DSPy threads).
>
> Future work should explore efficiency in a much broader evaluation to characterize this emerging space of LM programs. Just like the cost of training a new model, the cost of compiling a DSPy can vary greatly depending on the complexity of the program and the optimizer. However, we hope this small-scale evaluation sheds some light on the typical cost of compiling a program like this.
>
>
> >  There are no examples of compiled programs provided in the paper or appendices. [...] how do they differ from the types of prompt programs that people write by hand or using other frameworks? how do programs compiled for small LMs differ from those compiled for large LMs? are there any obvious patterns in the compiled programs?
>
> We now include a new appendix I (pages 28, 29, 30) which include snippets of our optimized prompts, when compiling with Llama2-13b-chat. We believe an extensive analysis of these prompts, especially when varying LMs and varying optimization strategies, is a highly exciting future direction.
>
> Nonetheless, it may be useful to consider Figure 11 (page 30) which reports on the Llama2-13b prompt for the second hop of query generation for HotPotQA. We see that it teaches the model two different strategies for generating search queries. The first bootstrapped few-shot example is a well-formed question "When was the first of the vampire-themed fantasy romance novels published?". The second is a keyword-based search query, "Jeremy Paxman birth year". It is possible to hypothesize this duality contributes to the selection of this prompt during optimization. Future work may try to characterize these patterns on a principled basis.
>
>
> > 'how about obvious shortcomings or irregularities, where additional hand-optimization would be easy?'
>
> One of the limitations of the BoostrapFewshot set of teleprompters we discuss in this work is that they focus on generating and selecting demonstrations of the input/output behavior of each module. To expand this toward more open-ended optimization, we have started multiple research directions that explore optimizing the instructions and execution logic (e.g., backtracking on failure) for the modules.
>
> [continue below...]

---

> ### Author Response · Authors · 2023-11-22
>
> [continued...]
>
>
> > 1. “In the abstract you say DSPy programs are "compiled to" a language model.”
> > 2. “If I understand correctly [teleprompters] are limited to gradient-free optimization techniques. This detail warrants mentioning.”
> > 3. “Typo: GMS08K -> GSM8K (page 2, paragraph 4) Typo: format -> formats (page 4, Section 3.2) Typo: grammar: but -> or (page 5, Section 3.3) Typo: "As we assumes" in the example at the end of Section 3.3 Typo: these -> this, ensembles -> ensembling (page 6, stage 3) Typo: "specific math problems or particular LM." -> "specific to math problems or a particular LM." (page 7)”
>
>
> Thank you for this extensive list of notes and typos. We believe we have now addressed all of these in the PDF. (We will update the submission abstract outside the PDF too, if the work is accepted. The system does not permit this change currently.)
>
>
>
> > Strong claim: The paper claims DSPy is the first programming model that translates prompting techniques into parameterized declarative modules that can be optimized. [...] Dohan 2022 is closest to the claim made in the paper.
>
>
> We agree to soften our statement in the introduction, in light of this note. In short, Dohan et al. (whose work we cite as early as our first paragraph) present a rich self-described “position paper” on the conceptual analogy between what they call “LM cascades” and probabilistic graphical models. While we find that work highly valuable, we are unable to immediately extrapolate from this conceptual analogy to a programming model in which prompting techniques can be optimized directly, as we do with DSPy. Nonetheless, to avoid any unintended implications, we have softened our statement in the introduction (marked in blue in page 2) in line with your comment.
>
>
> > Do the compilation approaches account for overfitting the validation set? [...] How do you think about safety during SQL generation, since SQL queries can modify or delete tables. How do you approach using tools with side effects during compilation?
>
> These are both good examples where using DSPy does not replace the value of good, thoughtful experimental design. In particular, care must be taken if the program has access to tools that can modify (or destroy) records or execute code, like running it in a sandbox. We have a small Python execution sandbox in DSPy to aid with this, but it is not an alternative for thoughtful design.
>
> Similarly, as with standard ML techniques, overfitting is an important consideration, especially when the training and validation sets are small. In our experience, we found limited signs of detrimental overfitting (i.e., selecting low-quality systems) but found evidence of sets of different difficulty such that the absolute scores of the systems are different across validation or evaluation sets.
>
> For good experimental design, the teleprompters allow us to distinguish between the training, validation, development, and test sets. For instance, when compiling the multi-hop program on HotPotQA, we use the first 50 examples of the training set for training, and examples 50 through 200 for validation. We compare different approaches on a separate development set (300 examples). And we finally report the quality on the test set (1000 separate examples). While the teleprompters enable such good experimental design, this cannot be enforced. It is always possible for misuse of the optimization process to overfit.
>
>
>
> > In the RAG example at the end of Section 3.2, is it complete without a definition of Retrieve provided? How can a DSPy user introduce a new Retrieve implementation?
>
> At the start of a script that uses DSPy, the user declares the “default” language model and retrieval model. These choices can also be changed for specific code blocks. In our experiments, we set the default retriever to be a ColBERTv2 (Santhanam et al., 2022) index over a Wikipedia corpus. However, this is very easy to extend. Our users have introduced many integrations for retrieval, from Pyserini to Pinecone to Weaviate, among other popular systems.

---

> > ### Comment · Reviewer_zPoD · 2023-11-22
> > **Thank you for your responses**
> >
> > Thank you for your responses to my review, and in particular for the multiple additions to the text (Appendixes A and I) that address my comments.
> >
> > While the addition of latency and throughput information that you've added in Appendix A is valuable for understanding the utility of DSPy, it does not address the main motivation I had for requesting that information; I was asking for some measure of cost in order to make the comparisons across programs and compilers in Tables 1 and 2 more meaningful. I would still encourage you to provide this.
> >
> > The addition of Appendix I does address my concern that no samples of generated programs were provided; however, I would still encourage you to provide some discussion of the qualitative aspects of these generated programs. There is no such discussion at present.
> >
> > Score-wise, your response has indeed increased my view of the paper (though not to the level of a "10", which would be required for me to adjust my score in OpenReview).

---

> ### Author Response · Authors · 2023-11-22
>
> Thank you for the thoughtful comments on our response. We are glad that you found our additions to the text valuable. We also appreciate that you considered raising the score (but couldn’t, because the system only accepts 10/10 as the next higher score).
>
>
> > some measure of cost in order to make the comparisons across programs and compilers in Tables 1 and 2 more meaningful
>
> We agree that readers will benefit from a comparison of the costs associated with different programs. In principle, measuring the compilation costs requires us to re-compile our programs (as we did for Appendix A, for one program) while disabling our caches. Considering that this is a significant experimental endeavor, if our work is accepted, we will include compiling and inference costs in the camera-ready version. This can potentially serve as a new column in the results tables or an expansion of Appendix A.
>
> We hope that the addition of Appendix A conveys the observation that the cost of compiling (which is a one-time, offline step) is fairly low. (As a ballpark, compiling is generally faster than typical finetuning jobs for instance.) Because of this, we typically recommend that DSPy users focus primarily on the quality and cost of the resulting programs, since compiling is an offline operation with manageable costs.
>
> Nonetheless, to provide a preliminary comparison of inference costs, we took four rows of increased complexity for HotPotQA (Table 2) and tested them with gpt-3.5-1106, while disabling the cache (which would otherwise skip any repeated computations, e.g. retrieval queries or LM calls).
>
> We ran 100 questions from HotPotQA with a single thread, and we report the average latency below. Each numbered bullet below represents a program (with compiler strategy in parentheses).
>
> **1. Vanilla (fewshot)**
>
> Average Latency: 0.3 seconds per question
>
> Average Cost: $0.0005 per question (1x)
>
> **2. CoT_RAG (fewshot)**
>
> Average Latency: 1.1 seconds per question
>
> Average Cost: $0.0013 per question (2.6x)
>
> **3. Multihop (fewshot)**
>
> Average Latency: 2.6 seconds per question
>
> Average Cost: $0.0018 per question (3.6x)
>
> **4. Multihop (bootstrap)**
>
> Average Latency: 2.6 seconds per question
>
> Average Cost: $0.0041 per question (8.2x)
>
> As this shows, these programs are within an order of magnitude of the cost and latency of the simplest one, even though we see major quality improvements from vanilla to multihop. We hope this provides deeper insight into our evaluations, and will expand on this analysis if our work is accepted.
>
>
> > I would still encourage you to provide some discussion of the qualitative aspects of these generated programs.
>
> We appreciate this suggestion. If the DSPy paper is accepted, we will use the time allocated for the camera ready (due in Feb) to expand Appendix I with a brief discussion of the prompts observed when compiling, and how that aids with debugging and analysis in our view, building off the feedback from (and discussion presented in our responses to) reviewers zPoD and cQiR.

---

### Official Review · Reviewer_FCUg · 2023-11-01

**Soundness:** 4 excellent
**Presentation:** 3 good
**Contribution:** 4 excellent
**Rating:** 8
**Confidence:** 3

**Summary:**

This paper introduces DSPy, a framework for expressing LM pipelines in a higher level language with support for automatic pipeline optimization. The results demonstrate that 1) complex pipelines can be written quickly and concisely in DSPy and 2) their automatic pipeline optimizations yield substantial gains in performance for two multi-step reasoning tasks (math reasoning and multi-hop QA).

**Strengths:**

DSPy is a major improvement over manually composing complex LM pipelines by hand. The paper also demonstrates the possibility of automatically optimizing parts of the pipeline once written in the DSPy framework. Finally, the experimental results are very impressive, particularly given the simplicity of user experience.

**Weaknesses:**

There is a major missing related work [1], which takes a similar approach of expressing LM pipelines as programs, and also comes with built-in optimizations. I would be happy to increase my score if the paper is revised to include a discussion comparing the two approaches.

A more minor concern is that the optimization techniques demonstrated here are relatively limited in scope. From a conceptual standpoint, I would have liked to see more than ensembling or bootstrapping few shot examples.

[1] Luca Beurer-Kellner, Marc Fischer, and Martin Vechev. 2023. Prompting Is Programming: A Query Language for Large Language Models. Proc. ACM Program. Lang. 7, PLDI, Article 186 (June 2023), 24 pages. https://doi.org/10.1145/3591300

====

The rebuttal has addressed my concerns and I have increased my score from a 6 to 8.

**Questions:**

Do the authors have any ideas for future optimization strategies (or "teleprompters") that could be implemented within this framework?

---

> ### Author Response · Authors · 2023-11-22
>
> Thank you for the helpful feedback. We are thrilled that you found our results 'very impressive' and that you appreciate the 'simplicity of the user experience' of the DSPy framework.
>
>
> > missing related work [1], which takes a similar approach of expressing LM pipelines as programs [...]  I would be happy to increase my score if the paper is revised to include a discussion comparing the two approaches.
>
> Thank you for pointing out LMQL. We have revised the related work section to compare LMQL and DSPy. We highlight this new part in blue. In short, LMQL is query language for efficiently constraining the decoding algorithms of LMs to generate only strings that fulfill logical constrains (e.g., lists of bullets or values formatted correctly for a calculator). LMQL’s lower-level interface for controlled decoding can be beneficial to implement specific advanced modules within DSPy.
>
> Given this, we believe that LMQL has fundamentally distinct design goals from DSPy. DSPy focuses on optimizing the quality of a pipeline of LM calls toward a given metric (e.g., optimizing the prompts of a multi-hop program to achieve high answer exact match). In contrast, LMQL helps make sure LM outputs adhere to specific *formatting* characteristics. The DSPy optimizations have an arbitrary continuous metric in mind, whereas LMQL optimizations focus on the efficiency of decoding under logical constraints.
>
> > A more minor concern is that the optimization techniques demonstrated here are relatively limited [...] Do the authors have any ideas for future optimization strategies (or "teleprompters") that could be implemented within this framework?
>
> Absolutely, we have a wide range of ideas for future optimization strategies. Our focus in this work is to show that the DSPy formulation of the problem (i.e., LM programs with optimizable modules, based on natural-language signatures) creates a very large space of potential optimizations. We then focus on showing that simple strategies for creating demonstrations (for prompting or finetuning), selecting between them, and ensembling them yield very large quality gains.
>
> Since we wrote this paper, we have also explored optimizing the instructions directly in a range of different ways, as well as strategies for automatic backtracking logic. We believe these extensions fall outside the scope of this paper, and we leave scientifically evaluating them for future work.
>
> ---
>
> Lastly, we appreciate your offer to raise the score given a discussion of LMQL vs DSPy. We believe this discussion of LMQL has helped us shed more light on the unique aspects of DSPy in the updated paper.

---

> > ### Comment · Reviewer_FCUg · 2023-11-22
> >
> > Thanks for the response. I do think the paper would be slightly stronger if there were some section covering future work, rather than leaving the potential of the framework up to the imagination of the reader.
> >
> > As my main concern regarding a comparison with LMQL is addressed, I have increased my score.

---

> > > ### Author Response · Authors · 2023-11-22
> > >
> > > We greatly appreciate your prompt feedback and the engagement with our discussion. We are glad that you raised the score for our work.
> > >
> > > > the paper would be slightly stronger if there were some section covering future work
> > >
> > > We agree with this remark. To this end, we added a new Appendix J (page 31), which includes a seven-paragraph (3/4 of a page) discussion of some of the future directions we see for DSPy. We have started exploring several of the directions mentioned there. If our work is accepted, we hope to be able to point to some of them in open-source, deanonymized form by the camera-ready deadline (in Feb 2024).
> > >
> > > While we list several promising directions in Appendix J, we elaborate in most depth on the creation (and evaluation) of a richer space of teleprompters (optimization and compilation strategies)  in DSPy. We hope the inclusion of this appendix strengthens the paper and benefits readers interested in the vision behind the DSPy framework.
> > >
> > > Thank you again for your feedback.

---

### Meta-Review · Area_Chair_gH4K · 2023-12-04

**Metareview:**

This work gives a reusable framework for building LLM pipelines, a timely topic, and refreshing in that the authors have taken seriously the goal of making a well-engineered software framework that could be generally applied across many problems. They also give a solid empirical evaluation of the work, and particularly show that they can tune smaller open models in their pipeline to be competitive with larger closed models. I am very happy to recommend acceptance of this work, preferably as a spotlight (at least).

**Justification For Why Not Higher Score:**

There are already LLM software frameworks, and the primary contribution of this work is not an interesting new algorithm, so it is not obvious that this work needs to be /aggressively/ highlighted as an oral (eg, pytorch was not even an academic publication, and this work is unlikely to be as impactful as pytorch)

**Justification For Why Not Lower Score:**

I think this work deserves to be highlighted as a spotlight: it stands to plausibly become widely used, and is refreshing to see this unconventional research style at this conference (developing programming models).

---

### Decision · Program_Chairs · 2024-01-16

Accept (spotlight)